# Pareto Merging:
# Multi-Objective Optimization for Preference-Aware Model Merging

**Weiyu Chen** [1]   **James T. Kwok** [1]

## Abstract

Model merging, which combines multiple models into a single model, has gained popularity in recent years. By efficiently integrating the capabilities of various models, this significantly reduces the parameter count and memory usage. However, current methods can only produce one single merged model. This necessitates a performance trade-off due to conflicts among the various models, and the resultant one-size-fits-all model may not align with the preferences of different users who may prioritize certain models over others. To address this issue, we propose preference-aware model merging, and formulate this as a multi-objective optimization problem in which the performance of the merged model on each base model's task is treated as an objective. In a single merging process, the proposed parameter-efficient structure generates a Pareto set of merged models, with each representing a Pareto-optimal solution for a preference. Users can then select merged models tailored to their preferences from this learned Pareto set. Experimental results demonstrate that the proposed Pareto Merging produces diverse trade-off models and achieves higher test accuracy compared to state-of-the-art merging baselines.

## 1. Introduction

Fine-tuning pre-trained models is a widely-adopted approach to create specialized models for various downstream tasks (He et al., 2022; Wortsman et al., 2022; Li et al., 2024a; Hu et al., 2021; Dettmers et al., 2024; Liu et al., 2024). Nowadays, many fine-tuned models are publicly available on online platforms such as HuggingFace. However, having

---
[1]Department of Computer Science and Engineering, The Hong Kong University of Science and Technology. Correspondence to: Weiyu Chen <wchenbx@cse.ust.hk>.

*Proceedings of the $42^{nd}$ International Conference on Machine Learning*, Vancouver, Canada. PMLR 267, 2025. Copyright 2025 by the author(s).

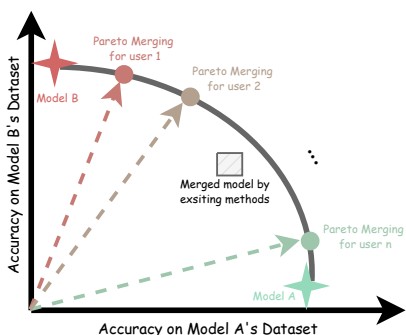

*Figure 1.* Comparison of existing methods and the proposed Pareto Merging for merging two models with varying user preferences.

multiple fine-tuned models results in a significant increase in parameter count and memory usage. As many of them originate from the same pre-trained model (e.g., ViT (Dosovitskiy et al., 2020)), it is thus desirable to combine them to a single model while still maintaining the full capabilities of these fine-tuned models, even without access to their original training data. This process, known as model merging (Matena & Raffel, 2022; Jin et al., 2023; Yang et al., 2024b), offers an efficient way to leverage the abundance of publicly accessible fine-tuned models.

A naive model merging approach is to average the weights of the fine-tuned models. However, this merged model often suffers from poor performance. To address this problem, various more advanced methods have been developed. For example, Matena & Raffel (2022) uses the Fisher matrix to preserve key features in the various models. RegMean (Jin et al., 2023) minimizes prediction discrepancies through weight adjustments. Ilharco et al. (2023) proposes to first obtain the differences (called *task vectors*) between the fine-tuned models and the original pre-trained model. A merged model is then created by combining the pre-trained model together with the sum of these task vectors. Further methods have been proposed to improve performance by reducing redundancy and conflicts among task vectors (Yadav et al., 2023), and addressing gradient mismatches (Daheim et al., 2024). Recently, AdaMerging (Yang et al., 2024b) is proposed that adaptively weights the task vectors based on the entropies of their model predictions.

An important limitation of existing model merging methods is that they can only produce a single merged model. Figure 1 shows an example of merging two fine-tuned models: model A on task A and model B on task B. Due to conflicts between the two models, typically the merged model (denoted by the square in the figure) cannot achieve the same accuracies on both tasks as models A and B. Hence, it needs to find a model with performance trade-off. However, this one-size-fits-all approach may not meet the diverse preferences of different users (shown as dotted vectors). For instance, user 1 may prefer to prioritize task B over task A, while user $n$ prioritizes task A over task B.

To overcome this limitation, we propose a preference-aware model merging approach, referred to as Pareto Merging (PM). This is formulated as a multi-objective optimization (MOO) problem (Miettinen, 1999) in which the performance on each model's task is considered as an objective. When there are $n$ user preferences, a naive integration of existing merging methods with MOO still requires storing the original fine-tuned models. Besides, it requires $n$ merging processes, which can be time expensive for merging methods that require optimization (e.g., AdaMerging). Thus, we introduce a novel parameter-efficient structure consisting of a preference-independent base model and a preference-dependent personalized model. In one single optimization process, PM learns the whole Pareto set of merged models, each representing the Pareto-optimal model for a user preference. From the learned Pareto set, one can provide merged models for different user preferences (denoted as circles in Figure 1). Moreover, PM can be used with model merging methods that do not require downstream data or only unlabeled data in the downstream task.

The main contributions of the paper are as follows:

- We formulate preference-aware model merging as multi-objective optimization, and provide diverse trade-off solutions according to user preferences.

- We introduce a preference-dependent personalized model using low-rank tensor, enabling diverse model generation with minimal parameter overhead.

- Experiments demonstrate that the proposed model outperforms existing state-of-the-art and can produce diverse models aligned with various user preferences.

## 2. Background

### 2.1. Model Merging

Model merging combines multiple deep networks to a single model. Existing methods can be divided into two categories: (i) merging models trained with different initializations (Ainsworth et al., 2023; Jordan et al., 2023; Stoica et al., 2024), and (ii) merging models fine-tuned on different datasets (Ilharco et al., 2023; Matena & Raffel, 2022; Jin et al., 2023; Yadav et al., 2023; Daheim et al., 2024; Yang et al., 2024b). In this paper, we focus on the latter.

Let the deep network be $f(\boldsymbol{x}; \boldsymbol{\theta}, \boldsymbol{h}_k)$, where $\boldsymbol{x}$ is the input data, $\boldsymbol{\theta}$ is the parameter of the shared bottom, and $\boldsymbol{h}_k$ is the task-specific head for the $k$th dataset. We denote the pretrained model's parameter as $\boldsymbol{\theta}_0$, and that after fine-tuning on dataset $k$ as $\boldsymbol{\theta}_k$. Since model merging focuses on the shared bottom, we use the simplified notation $f_{\boldsymbol{\theta}} \equiv f(\boldsymbol{x}; \boldsymbol{\theta})$.

The goal of model merging is to effectively combine $K$ fine-tuned models $f_{\boldsymbol{\theta}_1}, \ldots, f_{\boldsymbol{\theta}_K}$. A simple approach is weight averaging, but it often degrades performance. Fisher Merging (Matena & Raffel, 2022) uses the Fisher information matrix to ensure that crucial features from each task are preserved effectively. RegMean (Jin et al., 2023) minimizes the prediction differences between the merged model and individual models by adjusting their weights. Task Arithmetic (Ilharco et al., 2023) combines the task vectors $\{\boldsymbol{V}_k \equiv \boldsymbol{\theta}_k - \boldsymbol{\theta}_0\}_{k=1}^K$ to form a merged model $\boldsymbol{\theta}_0 + \lambda \sum_{k=1}^K \boldsymbol{V}_k$, where $\lambda \in \mathbb{R}$ controls the importance of task vectors. However, this might lead to sub-optimal performance due to potential conflicts among task vectors. To address this issue, TIES Merging (Yadav et al., 2023) designs operations to reduce redundancy and resolve sign conflicts among the task vectors. DARE (Yu et al., 2024) proposes to randomly drop some parameters and rescale the remaining parameters.

By observing that a lower Shannon entropy correlates well with model performance (Grandvalet & Bengio, 2004; Roy et al., 2022; Yang et al., 2024b), AdaMerging (Yang et al., 2024b) adaptively weighs the task vectors by minimizing the entropy averaged over all datasets. AdaMerging++ further pre-processes the task vectors with TIES Merging.

Several works enhance merging performance with task-specific modules but often add inference costs. Yang et al. (2024a) address representation distribution discrepancies between merged and unmerged models through representation surgery. Lu et al. (2024) propose dynamic merging using the mixture of experts (Cai et al., 2024), but it incurs extra inference costs and requires labeled data. EMR-Merging (Huang et al., 2024) uses task-specific masks with masking and rescaling during inference, introducing overhead. These methods are orthogonal to the proposed PM and can be combined with it (e.g., learning the Pareto set for these models). Notably, PM avoids extra inference costs compared to the base method once the user preference is specified.

Most related to the proposed method are rewarded soups (Rame et al., 2024) and MAP (Li et al., 2024b). The rewarded soups address multi-objective reinforcement learning with human feedback (RLHF) by training separate models for each objective and merging them through a simple weighted averaging of models based on preference vectors.

MAP employs an evolutionary MOO algorithm to search for the merging weights. A detailed comparison with these two approaches will be discussed in Section 3.3.

## 2.2. Multi-Objective Optimization

Multi-objective optimization (MOO) (Miettinen, 1999) aims to optimize $K$ objectives:

$$\min_{\boldsymbol{\mu}} \boldsymbol{g}(\boldsymbol{\mu}) = [g_1(\boldsymbol{\mu}), \ldots, g_K(\boldsymbol{\mu})]^\top, \tag{1}$$

where $\boldsymbol{\mu}$ is the variable and $g_k$ is the $k$th objective function. A solution $\boldsymbol{a}$ *dominates* another solution $\boldsymbol{b}$ if and only if $\forall k \in \{1, \ldots, K\}, g_k(\boldsymbol{a}) \leq g_k(\boldsymbol{b})$ and $\exists i \in 1, \ldots, K$ such that $g_i(\boldsymbol{a}) < g_i(\boldsymbol{b})$. A solution $\boldsymbol{a}$ *strictly dominates* another solution $\boldsymbol{b}$ if and only if $\forall k \in \{1, \ldots, K\}, g_k(\boldsymbol{a}) < g_k(\boldsymbol{b})$. A solution is *Pareto optimal* (resp. *weakly Pareto optimal*) if no other feasible solution *dominates* (resp. *strictly dominates*) it. The *Pareto set* is the set of all Pareto optimal solutions, while the *Pareto front* is the set of objective values of these Pareto optimal solutions.

A variety of gradient-based MOO algorithms have been developed and used in deep learning (Chen et al., 2025). Notable among these are MGDA (Sener & Koltun, 2018; Désidéri, 2012; Mukai, 1980; Fliege & Svaiter, 2000), EPO (Mahapatra & Rajan, 2020), CAGrad (Liu et al., 2021), Nash-MTL (Navon et al., 2022), and Auto-$\lambda$ (Liu et al., 2022). However, they can only output a single solution each run.

To better capture the Pareto set, several methods (Lin et al., 2019; Momma et al., 2022; Chen et al., 2024; Zhang et al., 2025) decompose the problem using a set of preference vectors, generating a single solution per run for each given preference. However, handling different preferences requires multiple runs, resulting in multiple models. To address this limitation and learn a continuous Pareto set, Pareto Hypernetworks (Navon et al., 2021; Lin et al., 2020; 2022) utilize a hypernetwork to generate network parameters based on a given preference vector. Preference-conditioned networks build on this by integrating preferences directly into the input layer (Ruchte & Grabocka, 2021) or through the FiLM layers (Dosovitskiy & Djolonga, 2020; Chen & Kwok, 2022). PaMaL (Dimitriadis et al., 2023) introduces a strategy that leverages the weighted sum of a set of base networks, while LORPMAN (Chen & Kwok, 2024) enhances efficiency using low-rank matrices. These approaches mainly focus on smaller models, such as the LeNet (LeCun et al., 1998) (30K parameters) or ResNet-18 (He et al., 2016) (11M parameters), and focus on training from scratch using large amounts of labeled data. In contrast, we explore model merging for large models (e.g., ViT (Dosovitskiy et al., 2020)) with limited unlabeled data or no data.

A detailed discussion on related works (Chen & Kwok, 2024; Dimitriadis et al., 2024) and concurrent works (Zhong et al., 2024; Tang et al., 2024) is in Appendix B.

## 3. Methodology

Despite the recent advances on model merging, a significant limitation is that they can only output a *single* trade-off model which cannot align with the diverse user preferences. To alleviate this problem, Section 3.1 first formulates model merging as a MOO problem. Section 3.2 introduces an efficient preference-dependent tensor structure. Finally, Section 3.3 shows how to solve the resultant optimization problem.

### 3.1. Model Merging as Multi-Objective Optimization

To accommodate different user preferences and thus achieve different trade-offs, we view model merging as a MOO problem. There are two common model merging scenarios.

#### 3.1.1. DATA-FREE MERGING

Recall that in Task Arithmetic (Ilharco et al., 2023), the merged model is $\boldsymbol{\theta} = \boldsymbol{\theta}_0 + \lambda \sum_{k=1}^{K} \boldsymbol{V}_k$, where $\lambda$ is a given hyperparameter. This can be interpreted as the closed-form solution of the optimization problem: $\min_{\boldsymbol{\theta}} \sum_{k=1}^{K} \|(\boldsymbol{\theta}_0 + \lambda K \boldsymbol{V}_k) - \boldsymbol{\theta}\|_F^2$, which minimizes the total distance of $\boldsymbol{\theta}$ to each scaled finetuned model $\boldsymbol{\theta}_0 + \lambda K \boldsymbol{V}_k$. However, it does not incorporate user preferences and is restricted to generating a single model.

We propose to reformulate model merging as the MOO problem with objective $S_k(\boldsymbol{\theta}) = \|(\boldsymbol{\theta}_0 + \lambda K \boldsymbol{V}_k) - \boldsymbol{\theta}\|_F^2$:

$$\min_{\boldsymbol{\theta}} [S_1(\boldsymbol{\theta}), \ldots, S_K(\boldsymbol{\theta})]^\top. \tag{2}$$

Given a preference vector $\boldsymbol{\gamma} = [\gamma_1, \ldots, \gamma_K]^\top$ in the simplex $\Delta^K \equiv \{\boldsymbol{\gamma} | \sum_{k=1}^{K} \gamma_k = 1, \gamma_k \geq 0\}$, a straightforward method to solve (2) is linear scalarization, which converts (2) to the single-objective optimization problem: $\min_{\boldsymbol{\theta}} \sum_{k=1}^{K} \gamma_k \|(\boldsymbol{\theta}_0 + \lambda K \boldsymbol{V}_k) - \boldsymbol{\theta}\|_F^2$. Its closed-form solution is:

$$\boldsymbol{\theta}(\gamma) = \boldsymbol{\theta}_0 + \lambda K \sum_{k=1}^{K} \gamma_k \boldsymbol{V}_k. \tag{3}$$

By varying $\boldsymbol{\gamma}$, different trade-offs can be achieved. Specifically, (1) When preferences are equal ($\boldsymbol{\gamma} = [1/K, \ldots, 1/K]^\top$), it recovers the solution of Task Arithmetic. (2) When $\lambda = 1/K$, the solution $\boldsymbol{\theta}_0 + \sum_{k=1}^{K} \gamma_k \boldsymbol{V}_k$ directly weights the task vectors with the preference vector, as in Rewarded Soups (Rame et al., 2024).

**Limitation.** Note that the solution subspace obtained in (3) is not parameter-efficient. It requires storing $KQ$ parameters[1], where $Q$ is the number of parameters in the

---

[1]Since the number of user preferences $n$ is usually larger than $K$, it is more space-efficient to store the $KQ$ task vectors and

pre-trained model. In other words, the numbers of stored parameters remains unchanged before and after merging.

### 3.1.2. MERGING BASED ON UNLABELED DATA

In AdaMerging (Yang et al., 2024b), instead of using a fixed $\lambda$ for all task vectors, a weighting vector $\boldsymbol{\lambda} = [\lambda_1, \ldots, \lambda_K]^\top$ is learned for each task vector by optimizing the entropy averaged over batches of *unlabeled* test data $\mathcal{B}_1, \ldots, \mathcal{B}_K$ sampled from $K$ datasets:

$$\min_{\boldsymbol{\lambda}} \sum_{k=1}^{K} \sum_{\boldsymbol{x} \in \mathcal{B}_k} H(f(\boldsymbol{x}; \boldsymbol{\theta}(\boldsymbol{\lambda}))) \text{ s.t.} \boldsymbol{\theta}(\boldsymbol{\lambda}) = \boldsymbol{\theta}_0 + \sum_{k=1}^{K} \lambda_k \boldsymbol{V}_k, \quad (4)$$

where $H(\cdot)$ is the Shannon entropy.[2] This approach demonstrates superior performance compared to data-free methods. However, it similarly lacks preference-awareness.

We propose to reformulate (4) to MOO by viewing the merged model's entropy $\sum_{\boldsymbol{x} \in \mathcal{B}_k} H(f(\boldsymbol{x}; \boldsymbol{\theta}))$ on each dataset $k$ as an objective $S_k(\boldsymbol{\theta})$:

$$\min_{\boldsymbol{\lambda}=[\lambda_1,\ldots,\lambda_K]^\top} [S_1(\boldsymbol{\theta}(\boldsymbol{\lambda})), \ldots, S_K(\boldsymbol{\theta}(\boldsymbol{\lambda}))]^\top$$

$$\text{s.t.} \qquad \boldsymbol{\theta}(\boldsymbol{\lambda}) = \boldsymbol{\theta}_0 + \sum_{k=1}^{K} \lambda_k \boldsymbol{V}_k. \quad (5)$$

Similar to data-free merging, given a user preference vector $\boldsymbol{\gamma}$, the naive MOO approach is linear scalarization:

$$\min_{\lambda} \sum_{k=1}^{K} \gamma_k S_k(\boldsymbol{\theta}(\boldsymbol{\lambda})) \text{ s.t. } \boldsymbol{\theta}(\boldsymbol{\lambda}) = \boldsymbol{\theta}_0 + \sum_{k=1}^{K} \lambda_k \boldsymbol{V}_k. \quad (6)$$

Different trade-off models can be obtained by varying $\boldsymbol{\gamma}$. In particular, with equal preferences, it recovers AdaMerging.

**Limitations.** For $n$ user preferences $\{\boldsymbol{\gamma}^{(1)}, \ldots, \boldsymbol{\gamma}^{(n)}\}$, $n$ runs of the optimization (6) are required to compute the corresponding $\boldsymbol{\lambda}^{(i)}$'s, which is computational inefficient. Furthermore, similar to data-free merging, one still requires storing $KQ$ parameters (the same as before merging).

> *Can we learn a parameter-efficient space of merged models in a single optimization process that can provide models tailored to varying user preferences?*

### 3.2. Parameter-Efficient Structure

To efficiently learn the Pareto set of preference-aware model merging, we propose a parameter-efficient structure with a

*preference-independent* base component and a *preference-dependent* personalized component. For the $l$th module in the backbone network (such as the attention module in a transformer), the base component includes the pretrained parameter $\boldsymbol{\theta}_0^l \in \mathbb{R}^{c^l \times d^l}$ and a weighted combination of task vectors $\sum_{k=1}^{K} \lambda_k^l \boldsymbol{V}_k^l$. The personalized component, parameterized as $\boldsymbol{W}^l(\boldsymbol{\gamma}) \in \mathbb{R}^{c^l \times d^l}$, depends on the user preference $\boldsymbol{\gamma}$. For simplicity, the description below focuses on a single module, omitting the superscript $l$.

Inspired by recent advancements in parameter-efficient fine-tuning (Hu et al., 2021; Dettmers et al., 2024; Liu et al., 2024), $\boldsymbol{W}(\boldsymbol{\gamma})$ has a low-rank structure. However, the popular approach of using a product of low-rank matrices as in LoRA (Hu et al., 2021) fails to account for user preferences. To provide a parameter-efficient way to integrate preference $\boldsymbol{\gamma}$, we model $\boldsymbol{W}(\boldsymbol{\gamma})$ as the following $c \times d \times 1$ tensor:[3]

$$\boldsymbol{W}(\boldsymbol{\gamma}) = \boldsymbol{G} \times_1 \boldsymbol{A} \times_2 \boldsymbol{B} \times_3 \boldsymbol{\gamma}, \quad (7)$$

where $\boldsymbol{G} \in \mathbb{R}^{r \times r \times K}$ is a core tensor, $\boldsymbol{A} \in \mathbb{R}^{r \times c}$, $\boldsymbol{B} \in \mathbb{R}^{r \times d}$ are matrices, and $r$ is a given rank parameter. Note that $\boldsymbol{E} \equiv \boldsymbol{G} \times_1 \boldsymbol{A} \times_2 \boldsymbol{B}$ is a $c \times d \times K$ tensor. This can be partitioned into $K$ slices $\boldsymbol{M}_1, \ldots, \boldsymbol{M}_K$, where each $\boldsymbol{M}_k \in \mathbb{R}^{c \times d}$ is low-rank and $[\boldsymbol{M}_k]_{i,j} = E_{i,j,k} = \sum_{q_1,q_2} G_{q_1,q_2,k} A_{q_1,i} B_{q_2,j}$. These matrices share parameters $\boldsymbol{A}$ and $\boldsymbol{B}$, while maintaining distinct characteristics through variations in $\boldsymbol{G}$. $\boldsymbol{W}(\boldsymbol{\gamma})$ in (7) is then a weighted sum of these low-rank matrices: $\boldsymbol{W}(\boldsymbol{\gamma}) = \sum_{k=1}^{K} \gamma_k \boldsymbol{M}_k$. By adjusting $\boldsymbol{\gamma}$, we modify the influence of each matrix to better align with user preference.

The whole model is then:

$$\boldsymbol{\theta}(\boldsymbol{\lambda}; \boldsymbol{\gamma}) = \boldsymbol{\theta}_0 + \sum_{k=1}^{K} \lambda_k \boldsymbol{V}_k + \boldsymbol{G} \times_1 \boldsymbol{A} \times_2 \boldsymbol{B} \times_3 \boldsymbol{\gamma}. \quad (8)$$

An illustration of the proposed structure is in Figure 2. Once the parameters $(\boldsymbol{G}, \boldsymbol{A}, \boldsymbol{B})$ are learned, an infinite number of models can be generated by simply varying $\boldsymbol{\gamma}$.

**Parameter Efficiency.** After merging, $\boldsymbol{\theta}_0 + \sum_{k=1}^{K} \lambda_k \boldsymbol{V}_k$ forms a single model with $cd$ parameters, matching the original pre-trained model. Thus, (8) adds only $r^2 K + r(c + d)$ parameters **after merging**. In the experiments, we apply the preference-dependent modification only to the transformer's attention modules. For example, with $r = 16$ in a ViT-B/32 model, the proposed approach increases parameters by just 0.5% compared to the pre-trained model.

---

compute merged models on-the-fly, rather than storing $n$ merged models ($nQ$ parameters).

[2] For a $M$-class classification problem, the Shannon entropy of prediction $\hat{\boldsymbol{y}} \in [0, 1]^M$ is $H(\hat{\boldsymbol{y}}) = -\sum_{m=1}^{M} \hat{y}_m \log \hat{y}_m$.

[3] $\times_u$ indicates the $u$-mode tensor product, which is detailed as follows: $(\boldsymbol{G} \times_1 \boldsymbol{A})_{q_1,i_2,i_3} = \sum_{i_1=1}^{r} G_{i_1,i_2,i_3} A_{i_1,q_1}$, for $q_1 = 1, \ldots, c$; $(\boldsymbol{G} \times_2 \boldsymbol{B})_{i_1,q_2,i_3} = \sum_{i_2=1}^{r} G_{i_1,i_2,i_3} B_{i_2,q_2}$, for $q_2 = 1, \ldots, d$; and $(\boldsymbol{G} \times_3 \boldsymbol{\gamma})_{i_1,i_2,q_3} = \sum_{i_3=1}^{m} G_{i_1,i_2,i_3} \gamma_{i_3}$, for $q_3 = 1$.

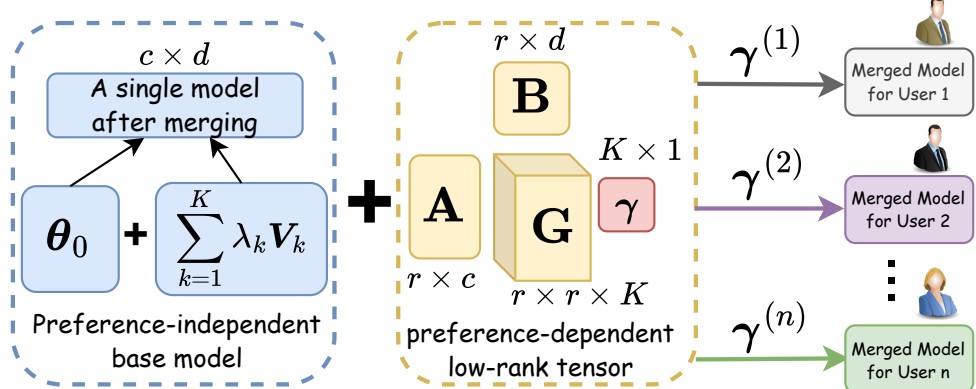

*Figure 2.* Illustration of the proposed Pareto Merging. After merging, it introduces minimal parameter overhead while providing different trade-off models to different user preferences.

### 3.3. Optimization

In this section, we consider the optimization problems for both data-free merging and unlabeled data-based merging. Since linear scalarization can only identify solutions on the convex parts of the Pareto front (Boyd & Vandenberghe, 2004), we use instead smooth Tchebycheff scalarization (Lin et al., 2024), which can identify the entire Pareto front. For the MOO problem in (1), smooth Tchebycheff scalarization considers the following optimization problem:

$$\min_{\boldsymbol{\mu}} \rho \log \left( \sum_{k=1}^{K} \exp \left( \frac{\gamma_k g_k(\boldsymbol{\mu}) - z_k^*}{\rho} \right) \right), \qquad (9)$$

where $\rho \in \mathbb{R}$ is a smoothing parameter, and $z_k^*$ is the ideal value of $g_k(\boldsymbol{\mu})$. It can be shown that $\boldsymbol{\mu}^*$ is a weakly Pareto optimal solution of the original MOO problem (1) if and only if it is also an optimal solution of (9) for some $\boldsymbol{\gamma}$.

In our context, we set $z_k^* = 0$, as the minimum of each $S_k(\boldsymbol{\theta})$ in either (2) (for data-free merging) or (5) (for unlabeled data-based merging) is zero.

We apply smooth Tchebycheff scalarization to problems (2) and (5) using the model in (8). The goal is to optimize the expected Tchebycheff scalarized objective value over a preference distribution $\mathcal{P}(\boldsymbol{\gamma})$:

$$\min_{\boldsymbol{G},\boldsymbol{A},\boldsymbol{B}} \quad \mathbb{E}_{\boldsymbol{\gamma}\sim\mathcal{P}(\boldsymbol{\gamma})} \rho \log \left( \sum_{k=1}^{K} \exp \left( \frac{\gamma_k S_k(\boldsymbol{\theta}(\boldsymbol{\lambda};\boldsymbol{\gamma}))}{\rho} \right) \right)$$
$$+ \beta \left\| \boldsymbol{G} \times_1 \boldsymbol{A} \times_2 \boldsymbol{B} \times_3 \boldsymbol{1}_K \right\|_F \qquad (10)$$
$$\text{s.t.} \quad \boldsymbol{\theta}(\boldsymbol{\lambda};\boldsymbol{\gamma}) = \boldsymbol{\theta}_0 + \sum_{k=1}^{K} \lambda_k \boldsymbol{V}_k$$
$$+ \boldsymbol{G} \times_1 \boldsymbol{A} \times_2 \boldsymbol{B} \times_3 \boldsymbol{\gamma}. \qquad (11)$$

For data-free approaches, $\boldsymbol{\lambda}$ is a user-provided hyperparameter. For merging with unlabeled data, we follow AdaMerging and optimize $\boldsymbol{\lambda}$ with the optimization objective (10). We

---

**Algorithm 1** Pareto Merging (PM).

preprocess the task vectors $\boldsymbol{V}_1, \ldots, \boldsymbol{V}_K$ (if required);
**while** not converged **do**
    sample $\boldsymbol{\gamma}$ from the Dirichlet distribution;
    obtain the corresponding model from (8);
    **if** data-free **then**
        $S_k(\boldsymbol{\theta}(\boldsymbol{\lambda};\boldsymbol{\gamma})) = \|\boldsymbol{\theta}_0 + \lambda K \boldsymbol{V}_k - \boldsymbol{\theta}(\boldsymbol{\lambda};\boldsymbol{\gamma})\|_F^2$;
    **else**
        */** using unlabeled data **/*
        sample a minibatch of *unlabeled* data $\boldsymbol{b}_1, \ldots, \boldsymbol{b}_K$;
        $S_k(\boldsymbol{\theta}(\boldsymbol{\lambda};\boldsymbol{\gamma})) = \sum_{\boldsymbol{x}\in\boldsymbol{b}_k} H(f(\boldsymbol{x};\boldsymbol{\theta}(\boldsymbol{\lambda};\boldsymbol{\gamma})))$;
    **end if**
    compute the objective in (11);
    update $\boldsymbol{G}, \boldsymbol{A}, \boldsymbol{B}$ (and $\boldsymbol{\lambda}$ if using unlabeled data) via gradient descent;
**end while**

---

adopt the symmetric Dirichlet distribution, $\text{Dir}([p, \ldots, p]^\top)$ with parameter $p$, for $\mathcal{P}(\boldsymbol{\gamma})$. By minimizing the expectation over the preference distribution, we aim to learn a model that minimizes the loss for each possible user preference (as the Dirichlet distribution spans the simplex of all possible user preferences). We also incorporate a regularizer in (10) to control the magnitude of model modification and prevent overfitting.

The proposed approach can be used with most merging techniques to enable preference-aware merging. As shown in Algorithm 1, a preference vector is sampled from the Dirichlet distribution in each iteration. For model merging using unlabeled data, we also sample a minibatch of unlabeled data $\boldsymbol{b}_1, \ldots, \boldsymbol{b}_K$ from $\mathcal{B}_1, \ldots, \mathcal{B}_K$. Next, we perform stochastic gradient descent on (11) to update the learnable tensor and matrices $\boldsymbol{G}, \boldsymbol{A}$, and $\boldsymbol{B}$. For model merging using unlabeled data, we also update the merging parameter $\boldsymbol{\lambda}$.

**Computation Complexity.** During training, the per-

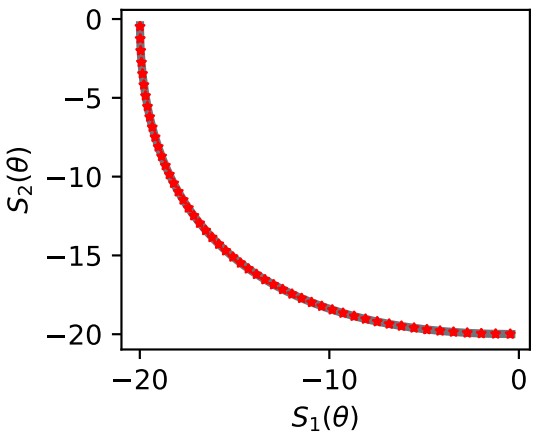

*Figure 3.* Solutions (red stars) sampled from the PF obtained by PM on the toy problem. The ground-truth PF is in gray.

iteration complexity of the proposed AdaMerging + PM is $O(Kcd + (c+d)r + Kr^2)$, whereas the original AdaMerging method has a complexity of $O(Kcd)$. Since $r$ is significantly smaller than $c$ and $d$, the additional computational overhead remains minimal. During inference, once the preference is fixed, the low-rank tensor can be merged into the base model, resulting in no additional inference overhead.

**Remark.** Rewarded Soups (Rame et al., 2024) applies preference-weighted merging to RLHF. In contrast to the proposed PM, it requires storing all task vectors. Moreover, the simple weighted averaging scheme it uses is often outperformed by more advanced data-free or data-based merging methods (Yadav et al., 2023; Yang et al., 2024b), as will be empirically shown in Section 4.2.1.

Similarly, the concurrent work MAP (Li et al., 2024b) again requires storing all task vectors to create trade-off models. Additionally, it can only produce a discrete approximation of the Pareto set. In contrast, the proposed PM stores only a single merged model and a small low-rank tensor, providing a continuous approximation of the Pareto set and can directly generate models for any preference.

## 4. Experiments

In this section, we perform experiments on both toy problem (Section 4.1) and real-world datasets (Section 4.2). Ablation study is provided in Section 4.3.

### 4.1. Toy Problem

We first demonstrate that the proposed algorithm can learn the Pareto set by using a popular toy problem (Liu et al., 2021; Navon et al., 2022) with two objectives $S_1(\boldsymbol{\theta})$ and $S_2(\boldsymbol{\theta})$. Due to the limited space, the detailed definition can be found in Appendix A.

We first obtain $\boldsymbol{\theta}_1$ by optimizing $S_1$, and $\boldsymbol{\theta}_2$ by optimizing $S_2$. The average $(\boldsymbol{\theta}_1 + \boldsymbol{\theta}_2)/2$ serves as the preference-independent base model, which is then fixed. Subsequently, we optimize $\boldsymbol{G}$, $\boldsymbol{A}$, and $\boldsymbol{B}$ in the preference-dependent model. Figure 3 shows models sampled from the learned Pareto set using 31 uniformly distributed preferences. As can be seen, the solutions overlap with the ground-truth PF and uniformly cover the entire PF.

### 4.2. Real-World Datasets

Following (Ilharco et al., 2023; Yang et al., 2024b), we use the vision encoders (ViT-B/32 and ViT-L/14) from CLIP (Radford et al., 2021) as pre-trained backbones. Unless otherwise specified, ViT-B/32 is used. This is fine-tuned on the following eight image classification datasets as in (Ilharco et al., 2023; Yang et al., 2024b): SUN397, Cars, RESISC45, EuroSAT, SVHN, GTSRB, MNIST, and DTD. Details can be found in Appendix C.

#### 4.2.1. MERGING TWO MODELS

In this experiment, we merge the two models fine-tuned on RESISC45 and GTSRB. Pareto Merging is applied to the baselines of Task Arithmetic (data-free) and AdaMerging (with unlabeled data). We also compare with two baselines that can obtain the Pareto set: Rewarded Soups (Rame et al., 2024) (data-free) and MAP (Li et al., 2024b) (with data). We follow MAP's official implementation that uses labeled data and a population size of 100.

Figure 4 shows the test accuracies of merged models across 11 preference vectors (from $[0.0, 1.0]^\top$ to $[1.0, 0.0]^\top$). For clarity, only a subset of the solutions is shown. The complete plot is shown in Appendix D.1. As can be seen, PM, combined with baseline methods, effectively generates diverse trade-off models across the two datasets, enabling users to choose models that align with their preferences. For example, with a preference vector of $\gamma = [1.0, 0.0]^\top$, which prioritizes RESISC45 exclusively, the model generated by AdaMerging + PM (the rightmost green star in Figure 4) achieves an accuracy of 94.8% on RESISC45, while the original AdaMerging method (without preference consideration) achieves only 93.6% accuracy on RESISC45. Similarly, Task Arithmetic + PM (the rightmost blue star) achieves 92.3% accuracy on RESISC45, while the original Task Arithmetic achieves 91.2% accuracy. These clearly demonstrate the advantages of preference-aware Pareto Merging over one-size-fits-all models.

Compared to the rewarded soups, Task Arithmetic + PM (which is also data-free) achieves a much better trade-off. Similarly, AdaMerging + PM outperforms MAP. Notably, while MAP relies on labeled data (which may not be feasible in practice), PM only requires unlabeled data. Due to rewarded soups' significantly inferior performance, it is

*Table 1.* Comparison of training time (on an NVIDIA A6000), parameter count **after merging** and the number of models that one can obtain after merging two models.

| Method | GPU hours | # params | # models |
|---|---|---|---|
| Rewarded Soups | $\approx 0$ | 226.7M | infinite |
| Task Arithmetic | $\approx 0$ | 113.4M | 1 |
| Task Arithmetic + PM | 0.04 | 114.0M | infinite |
| MAP | 0.35 | 226.7M | $\approx 500$ |
| AdaMerging | 0.15 | 113.4M | 1 |
| AdaMerging + PM | 0.28 | 114.0M | infinite |

excluded from further comparisons.

Table 1 compares the training time, parameter count after merging, and number of models generated. As can be seen, while PM requires slightly more time than the baselines, it adds minimal parameter overhead after merging and can generate an infinite number of trade-off models. When compared to Rewarded Soups and MAP, after merging, both Rewarded Soups and MAP result in a parameter count that is double that of the pre-trained model, as they require storing all the original models. This overhead becomes even more significant as $K$ increases.

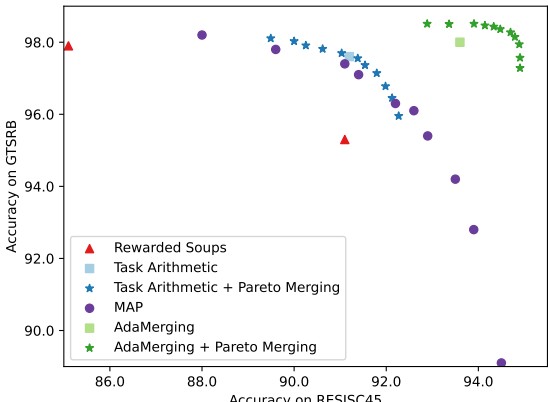

*Figure 4.* Accuracies of models obtained by different methods when merging two ViT-B/32 models.

We also compare PM with two straightforward methods discussed in Section 3.1: Task Arithmetic + Preference (Section 3.1.1) and AdaMerging + Preference (Section 3.1.2). The results are shown in Figure 5. For clarity, only a subset of the solutions is shown. The complete plot is shown in Appendix D.1.

As shown in Figures 5, Task Arithmetic + Preference has a large parameter overhead of 113.4M and some of the extreme models it obtained are typically undesirable for users. In contrast, Task Arithmetic + PM with its default configuration (applying preference-dependent modification

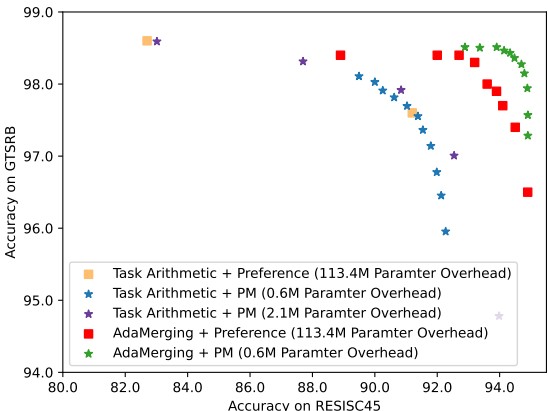

*Figure 5.* Comparison with two straightforward methods discussed in Section 3.1 when merging two ViT-B/32 models.

to attention layers), achieves a reasonable trade-off with a significantly reduced parameter overhead of only 0.6M. Furthermore, extending preference-dependent modifications to both the attention and MLP layers results in a broader spread of PF, while keeping the parameter overhead modest at 2.1M.

A similar advantage is observed with the proposed AdaMerging + PM method. It achieves a superior trade-off with 0.6M parameter overhead, compared to the 113.4M overhead incurred by AdaMerging + Preference. Beyond parameter efficiency, AdaMerging + PM also offers significant computational savings. For instance, AdaMerging + PM requires only a single execution run (0.28 GPU hours), whereas AdaMerging + Preference requires 11 runs (1.65 GPU hours) to handle 11 preference vectors.

### 4.2.2. MERGING EIGHT MODELS

In this experiment, we utilize all eight models. The proposed method PM is applied to two representative data-free merging approaches: Task Arithmetic and TIES-Merging (Yadav et al., 2023). Additionally, we apply PM to AdaMerging and its enhanced version AdaMerging++ (to avoid confusion, this is referred to as AdaMerging+TIES in the sequel). We also compare with other baselines, including Weight Averaging (Yadav et al., 2023; Yang et al., 2024b), Fisher Merging (Matena & Raffel, 2022), RegMean (Jin et al., 2023), DARE (Yu et al., 2024), and MAP. For MAP, We use the nested merging variant in the official code.

Figure 6 shows the accuracies of 30 models sampled from the learned Pareto set by AdaMerging + PM. Since the Pareto front with more than three objectives cannot be directly visualized, each subplot in Figure 6 shows the accuracies on 2 of the 8 tasks. Results on the other baselines +

*Table 2.* Test accuracies when merging eight ViT-B/32 models. Results on DARE and MAP are reproduced using official implementations. Results on the remaining baselines are from (Yang et al., 2024b). The best result among each group is in bold.

| Method | SUN397 | Cars | RESISC45 | EuroSAT | SVHN | GTSRB | MNIST | DTD | **Average** |
|---|---|---|---|---|---|---|---|---|---|
| Individual | 75.3 | 77.7 | 96.1 | 99.7 | 97.5 | 98.7 | 99.7 | 79.4 | 90.5 |
| Traditional MTL | 73.9 | 74.4 | 93.9 | 98.2 | 95.8 | 98.9 | 99.5 | 77.9 | 88.9 |
| Weight Averaging | 65.3 | 63.4 | 71.4 | 71.7 | 64.2 | 52.8 | 87.5 | 50.1 | 65.8 |
| Fisher Merging | 68.6 | 69.2 | 70.7 | 66.4 | 72.9 | 51.1 | 87.9 | 59.9 | 68.3 |
| RegMean | 65.3 | 63.5 | 75.6 | 78.6 | 78.1 | 67.4 | 93.7 | 52.0 | 71.8 |
| DARE | 54.8 | 54.6 | 66.6 | 78.3 | 80.2 | 69.8 | 97.3 | 49.8 | 68.9 |
| MAP | 60.0 | 58.8 | 85.8 | 69.5 | 83.5 | 73.4 | 87.8 | 53.2 | 71.5 |
| Task Arithmetic | 55.2 | 54.9 | 66.7 | 78.9 | 80.2 | 69.7 | 97.3 | 50.4 | 69.1 |
| Task Arithmetic+PM (equal) | 55.2 | 55.0 | 66.7 | 78.9 | 80.2 | 69.7 | 97.3 | 50.4 | 69.2 |
| Task Arithmetic+PM (priority) | **55.5** | **55.9** | **67.6** | **81.8** | **84.5** | **73.3** | **97.9** | **51.5** | **71.0** |
| TIES-Merging | 59.5 | 60.0 | 71.7 | 78.2 | 86.3 | 72.9 | 98.2 | 52.8 | 72.4 |
| TIES-Merging+PM (equal) | 59.5 | 60.0 | 71.8 | 78.3 | 86.3 | 72.9 | 98.2 | 52.8 | 72.4 |
| TIES-Merging+PM (priority) | **60.0** | **60.6** | **72.9** | **81.7** | **89.2** | **75.8** | **98.4** | **53.8** | **74.1** |
| AdaMerging | 64.5 | 68.1 | 79.2 | 93.8 | 87.0 | 91.9 | 97.5 | 59.1 | 80.1 |
| AdaMerging+PM (equal) | 70.1 | **74.2** | 87.3 | 96.5 | 90.2 | 95.6 | 98.5 | **66.7** | 84.9 |
| AdaMerging+PM (priority) | **71.1** | **74.2** | **89.0** | **97.6** | **92.1** | **97.4** | **99.0** | 64.0 | **85.5** |
| AdaMerging+TIES | 66.6 | 68.3 | 82.2 | 94.2 | 89.6 | 89.0 | 98.3 | 60.6 | 81.1 |
| AdaMerging+TIES+PM (equal) | 70.6 | **73.9** | 87.5 | 96.7 | 90.8 | 96.7 | 98.6 | **67.2** | 85.2 |
| AdaMerging+TIES+PM (priority) | **72.1** | 73.7 | **88.8** | **97.5** | **92.2** | **97.5** | **99.0** | 66.1 | **85.9** |

PM are in Appendix D.2. As can be seen, PM again offers good and diverse trade-off solutions.

Table 2 shows the test accuracies of Pareto Merging combined with four baselines, along with various others. We also compare with individual task learning and traditional multi-task learning (MTL), which serve as performance upper bounds and require labeled training data and are *significantly more computationally expensive*. Moreover, as showing results for all users preferences is infeasible, we only show two representative types of preferences: (1) equal preference: $\gamma = [0.125, \ldots, 0.125]^\top$; and (2) priority preference, in which one task has a weight of $0.5$, while each remaining task has a weight of $0.5/(K-1)$. This reflects the common scenario where users prioritize a specific task while still expecting reasonable performance on the others.

Since the nested merging variant of MAP is used when merging more than two models, multiple runs are required to incorporate different user preferences. Thus, we only present the MAP results with equal preference.

As can be seen, for Task Arithmetic and TIES-Merging, Pareto Merging with priority preference significantly improves performance. For example, on SVHN, the accuracy increases by 4.3% and 2.9% for Task Arithmetic and TIES-Merging, respectively. Additionally, the average accuracy increases by 1.9% and 1.7%, respectively. This clearly demonstrates that PM can efficiently incorporate preferences to provide better performance for data-free method.

For AdaMerging and AdaMerging+TIES, Pareto Merging shows even larger performance gains with the help of unlabeled data. Even with equal preference, PM achieves an average performance improvement of 4.8% and 4.1%, respectively. With priority preference, the performance gain is even more pronounced. For instance, on RESISC45, AdaMerging+PM (priority) outperforms AdaMerging by 10%. These demonstrate that PM can efficiently incorporate preferences to achieve better performance for unlabeled-data based methods. We further experiment on the larger ViT-L/14 backbone. Results can be found in Appendix E.

### 4.2.3. UNSEEN DATASETS

In this experiment, we assess the performance on unseen datasets with no fine-tuned model. Following (Yang et al., 2024b), we consider two settings: (i) Merge models fine-tuned on SUN397, Cars, RESISC45, SVHN, GTSRB, and DTD, and then evaluate on MNIST and EuroSAT. (ii) Merge models fine-tuned on SUN397, Cars, EuroSAT, GTSRB, MNIST, and DTD, and then evaluate on RESISC45 and SVHN. For Pareto Merging, we randomly sample 30 models from the learned Pareto set, and select the one with the smallest Shannon entropy on the target dataset.

Test accuracies are shown in Tables 3 and 4. As can be seen, combining with Pareto Merging results in higher accuracies compared to using the base methods alone. This demonstrates an important advantage of learning the Pareto set, namely that we can select the most suitable model from the

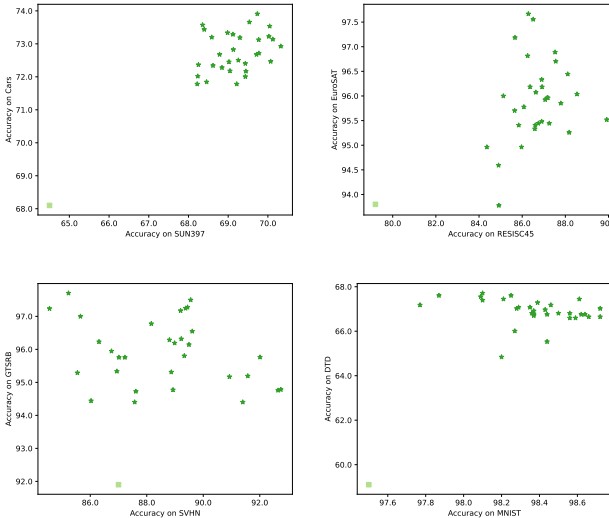

*Figure 6.* Models sampled from the learned Pareto set by AdaMerging + PM when merging 8 ViT-B/32 models. Each subplot shows the accuracies on 2 of the 8 tasks. For comparison, the square denotes the model obtained by AdaMerging.

*Table 3.* Test accuracies on unseen MNIST and EuroSAT when merging six ViT-B/32 models. Results on baselines are from (Yang et al., 2024b).

| Method | MNIST | EuroSAT | **Average** |
|---|---|---|---|
| Task Arithmetic | 77.2 | 46.2 | 61.7 |
| Task Arithmetic+PM | **79.3** | **46.8** | **63.0** |
| TIES Merging | 75.9 | 43.3 | 59.6 |
| TIES Merging+PM | **80.7** | **43.4** | **62.1** |
| AdaMerging | 84.0 | 56.1 | 70.0 |
| AdaMerging+PM | **84.3** | **65.1** | **74.7** |
| AdaMerging+TIES | 83.9 | 53.5 | 68.7 |
| AdaMerging+TIES+PM | **85.9** | **65.4** | **75.7** |

set, while the base methods do not have this freedom.

### 4.3. Ablation Studies

In this section, we perform ablation studies on (i) the rank $r$ and (ii) using outer product instead of tensor-based model in (7). Because of the lack of space, results on the outer product model can be found in Appendix F.

The default hyperparameter settings are the same as in Section 4.2. Experiments are performed by using AdaMerging + PM on the two models fine-tuned on RESISC45 and GT-SRB. For each setting, we uniformly sample 11 models from the learned Pareto set.

Figure 7 shows the test accuracies with different ranks ($r$). As can be seen, increasing $r$ improves model performance in general, as a higher rank allows the model greater flexibility to adapt to various preferences. However, a very large rank

*Table 4.* Test accuracies on unseen RESISC45 and SVHN when merging six ViT-B/32 models. Results on baselines are from (Yang et al., 2024b).

| Method | RESISC45 | SVHN | **Average** |
|---|---|---|---|
| Task Arithmetic | 52.3 | 49.9 | 51.1 |
| Task Arithmetic+PM | **52.4** | **51.9** | **52.1** |
| Ties Merging | 58.7 | 49.2 | 53.9 |
| TIES Merging+PM | **59.0** | **51.9** | **55.5** |
| AdaMerging | 50.2 | 60.9 | 55.5 |
| AdaMerging+PM | **52.4** | **62.6** | **57.5** |
| AdaMerging+TIES | 52.0 | **64.9** | 58.5 |
| AdaMerging+TIES+PM | **52.9** | 64.7 | **58.8** |

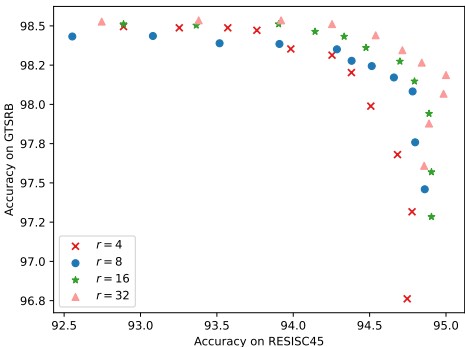

*Figure 7.* Effects of rank $r$.

may leads to overfitting. Additionally, a higher rank also introduces more parameters,[4] though it is still minimal.

## 5. Conclusion

In this paper, we address the challenge of model merging through the lens of multi-objective optimization. We introduce Pareto Merging, which enables the generation of an *infinite* number of Pareto-optimal models based on user preferences with a *single* merging process and minimal parameter overhead. Experimental results demonstrate that PM can generate different trade-off solutions, better aligning with user preferences compared to non-preference-aware baselines. In the future, we will consider extending the experiment to large language models.

## Acknowledgment

This research was supported in part by the Research Grants Council of the Hong Kong Special Administrative Region (Grant 16202523 and HKU C7004-22G).

---

[4]For $r = 4/8/16/32$, the total number of parameters is increased by $0.1\%/0.3\%/0.5\%/1.0\%$ compared to the pre-trained model.

## Impact Statement

This paper aims to advance the field of Machine Learning. There are many potential societal consequences of our work, none of which must be specifically highlighted here.

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

## A. Details of the Toy Problem

We use a popular toy problem (Liu et al., 2021; Navon et al., 2022) with two objectives $S_1(\boldsymbol{\theta})$ and $S_2(\boldsymbol{\theta})$. The original problem has a parameter $\boldsymbol{\theta} \in \mathbb{R}^2$. To create a problem with a matrix-form parameter, we slightly modify it to $\boldsymbol{\theta} \in \mathbb{R}^{6 \times 6}$. The problem is then reformulated as follows:

$$\text{Minimize } S_1(\boldsymbol{\theta}) = c_1(\boldsymbol{\theta})h_1(\boldsymbol{\theta}) + c_2(\boldsymbol{\theta})g_1(\boldsymbol{\theta}) \text{ and } S_2(\boldsymbol{\theta}) = c_1(\boldsymbol{\theta})h_2(\boldsymbol{\theta}) + c_2(\boldsymbol{\theta})g_2(\boldsymbol{\theta}),$$

$$\text{where } \theta^{(1)} = \sum_{i=1}^{3}\sum_{j=1}^{6}\theta_{i,j}, \text{ and } \theta^{(2)} = \sum_{i=4}^{6}\sum_{j=1}^{6}\theta_{i,j},$$

$$h_1(\boldsymbol{\theta}) = \log\left(\max(|0.5(-\theta^{(1)} - 7) - \tanh(-\theta^{(2)})|, \ 0.000005)\right) + 6,$$

$$h_2(\boldsymbol{\theta}) = \log\left(\max(|0.5(-\theta^{(1)} + 3) - \tanh(-\theta^{(2)}) + 2|, \ 0.000005)\right) + 6,$$

$$g_1(\boldsymbol{\theta}) = \left((-\theta^{(1)} + 7)^2 + 0.1 * (-\theta^{(2)} - 8)^2\right)/10 - 20,$$

$$g_2(\boldsymbol{\theta}) = \left((-\theta^{(1)} - 7)^2 + 0.1 * (-\theta^{(2)} - 8)^2\right)/10 - 20,$$

$$c_1(\boldsymbol{\theta}) = \max(\tanh(0.5 * \theta^{(2)}), \ 0) \text{ and } c_2(\boldsymbol{\theta}) = \max(\tanh(-0.5 * \theta^{(2)}), \ 0).$$

We first obtain $\boldsymbol{\theta}_1$ by optimizing $S_1$, and $\boldsymbol{\theta}_2$ by optimizing $S_2$. The average $(\boldsymbol{\theta}_1 + \boldsymbol{\theta}_2)/2$ serves as the preference-independent base model, which is then fixed. Subsequently, we optimize $\boldsymbol{G}$, $\boldsymbol{A}$, and $\boldsymbol{B}$ in the preference-dependent model.

The rank $r$ is set to 2. We use the Adam optimizer with a learning rate of 0.001. We use 31 uniformly distributed preference vectors (i.e., $[1, 0]^\top, [1/30, 29/30]^\top, ..., [0, 1]^\top$). We can see that the solutions uniformly cover the entire PF in the objective space.

## B. Detailed Discussion of some Related Works

This section provides a detailed comparison of the proposed approach with several related works, highlighting the key differences and advantages of our method.

**Comparison with Pareto Set Learning using Low-Rank Structures.** While works like (Chen & Kwok, 2024; Dimitriadis et al., 2024) use low-rank structures for Pareto set learning, our primary aim is to improve model merging itself, not use merging for Pareto set learning. We reformulate model merging as a MOO problem to integrate user preferences and propose an efficient preference-aware tensor structure, different from the weighted LoRA sums in (Chen & Kwok, 2024; Dimitriadis et al., 2024). To demonstrate the advantage of the proposed, we adapted (Chen & Kwok, 2024; Dimitriadis et al., 2024) for model merging (eight ViT-B/32 models, setup from Section 4.2.2, Table 2). Results are in Table 5.

*Table 5.* Comparison with low-rank structures from Pareto set learning (Chen & Kwok, 2024; Dimitriadis et al., 2024) on merging eight ViT-B/32 models.

| Method | Structure | Test Accuracy (%) | Parameter Overhead (M) |
|---|---|---|---|
| AdaMerging+PM (equal) | Ours | 84.9 | 0.61 |
| AdaMerging+PM (equal) | Structure from (Chen & Kwok, 2024; Dimitriadis et al., 2024) | 84.5 | 4.71 |
| AdaMerging+PM (priority) | Ours | 85.5 | 0.61 |
| AdaMerging+PM (priority) | Structure from (Chen & Kwok, 2024; Dimitriadis et al., 2024) | 85.2 | 4.71 |

The results in Table 5 show that our proposed tensor structure achieves superior performance while substantially reducing the number of additional parameters compared to adapting structures from (Chen & Kwok, 2024; Dimitriadis et al., 2024) to the model merging task.

**Comparison with concurrent LLM Alignment Work.** The proposed method differs from the concurrent LLM alignment work Panacea (Zhong et al., 2024) in three main aspects: (1) Problem Formulation: Panacea (Zhong et al., 2024) targets mutli-objective LLM alignment, while we formulated model merging (data-free and data-based) as MOO, enabling preference-driven model generation and improved performance. (2) Low-Rank Structure: Panacea (Zhong et al., 2024) uses an SVD-LoRA approach. Our novel low-rank tensor structure is more flexible, adaptively learning inter-objective

relationships. Table 6 compares our structure with Panacea (Zhong et al., 2024) (rank 16 for comparable parameters) on merging eight ViT-B/32 models. Our structure outperforms Panacea, especially with priority preferences. (3) Optimization & Data: Panacea (Zhong et al., 2024) uses labeled data for LLM alignment. We focus on model merging with limited (unlabeled) data or no data, addressing potential overfitting via our efficient structure and tensor regularization.

*Table 6.* Comparison with SVD-LoRA structure from (Zhong et al., 2024) on merging eight ViT-B/32 models.

| Method | Model Structure | Test Accuracy (%) |
|---|---|---|
| AdaMerging+PM (equal) | Ours | 84.9 |
| AdaMerging+PM (equal) | Structure from Panacea (Zhong et al., 2024) | 84.1 |
| AdaMerging+PM (priority) | Ours | 85.5 |
| AdaMerging+PM (priority) | Structure from Panacea (Zhong et al., 2024) | 84.4 |

**Comparison with Concurrent Pareto Set Learning via MoE.** Another relevant concurrent work (Tang et al., 2024) focuses on enhancing the efficiency of general Pareto set learning algorithms, a goal different from our primary objective of advancing model merging through MOO. Tang et al. (2024) employs task arithmetic to merge selected modules from pre-trained models while leaving other components (e.g., MLP layers, or both MLP and attention layers) separate. A Mixture-of-Experts (MoE) router is then trained using labeled data to assign weights to these unmerged parts. On the other hand, our method merges all modules into a single model with a small low-rank tensor (approximately 0.5% of pre-trained model parameters) and does not require labeled data.

To further compare with MoE-based fusion in (Tang et al., 2024), below we compare its performance on merging eight ViT-B/32 models. As reported in (Tang et al., 2024), with the use of labeled data, MoE-based fusion (with a final model size of 567M) achieves an average accuracy of 77.2% when only the MLP is unmerged, and 83.5% when all modules are unmerged (with final model size 1.02B). On the other hand, our method (with only 114M parameters and not requiring labeled data) achieves a higher accuracy at 85.2%.

## C. Experimental Details

### C.1. Datasets

The datasets used are summarized in Table 7. All datasets are publicly available. The Cars dataset has a custom license restricted to non-commercial use. The DTD dataset has a custom license restricted to research-only use. EuroSAT is under the MIT license. The licenses for the remaining datasets are unknown.

*Table 7.* Summary of the datasets used.

| Dataset | Domain | # Classes | # Images |
|---|---|---|---|
| SUN397 (Xiao et al., 2016) | Scene classification | 397 | 108,754 |
| Cars (Krause et al., 2013) | Car classification | 196 | 16,185 |
| RESISC45 (Cheng et al., 2017) | Remote sensing scene classification classification | 45 | 31,500 |
| EuroSAT (Helber et al., 2019) | Satellite image classification classification | 10 | 27,000 |
| SVHN (Netzer et al., 2011) | House numbers classification | 10 | 600,000 |
| GTSRB (Stallkamp et al., 2011) | Traffic sign classification | 43 | 51,839 |
| MNIST (LeCun, 1998) | Handwritten digits classification | 10 | 70,000 |
| DTD (Cimpoi et al., 2014) | Texture classification | 47 | 5,640 |

### C.2. Training Details

We use the checkpoints of pre-trained and fine-tuned models on eight datasets in (Ilharco et al., 2023). Following (Yang et al., 2024b), we initialize all $\{\lambda_k\}_{k=1}^K$ in (8) to 0.3. We employ the Adam optimizer (Kingma & Ba, 2014), with learning rate $1 \times 10^{-3}$ and momentum parameters $\beta_1, \beta_2$ set to 0.9 and 0.999, respectively. The batch size is 32. We set the number of optimization steps to 2000 when merging 2 models and 4000 when merging 8 models. We initialize $G$ in (8) to the zero

tensor, and initialize $A$ and $B$ using the Kaiming uniform distribution (He et al., 2015). We set rank $r$ to 16, and both the regularization coefficient $\beta$ and distribution parameter $p$ to 0.1.

For the baseline methods, we use the hyperparameters provided in the original papers and adopt their publicly available official implementations. Specifically, for Task Arithmetic, we set $\lambda$ to 0.3 when merging eight models, as suggested in the original paper. When merging two models, we experiment with $\lambda$ values from $\{0.1, 0.3, 0.5, 0.7, 0.9\}$ and select $\lambda = 0.7$, which achieves the best performance on the validation dataset. For experiments on the ViT-B/32 (resp. ViT-L/14) model, we use a single NVIDIA A6000 (resp. H800) GPU with 48GB (resp. 80GB) memory. We use Ubuntu 22.04.1 with PyTorch 1.12.

## D. Additional Figures of Models Obtained by Pareto Merging

### D.1. Complete Plot When Merging Two Models

Figure 8 shows the complete plot when merging two models. Rewarded soups can achieve very diverse solutions, as they retain all original models. However, their performance in the middle region is notably poor. MAP partially improves performance in this region through evolutionary optimization with labeled data, but it still falls significantly short compared to AdaMerging + PM. Most solutions obtained by Rewarded soups and AdaMerging + PM are dominated by solutions obtained by AdaMerging + PM, except some extreme solutions. Note that, in general, users are more likely to prefer a model that performs well on priority tasks while maintaining satisfactory performance on other tasks. It is unlikely that users would select an extreme solution with slightly improvement on a task (e.g, from 95.8% to 95.9%) while causing a significant performance drop in another task (e.g., from 52.1% to 22.4%).

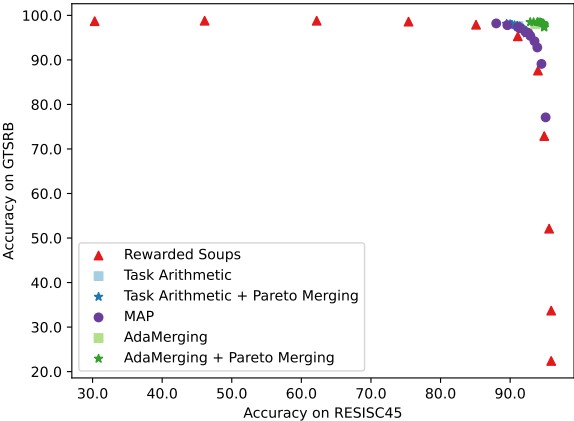

*Figure 8.* Full plot of accuracies of models obtained by different methods when merging two ViT-B/32 models.

### D.2. Figures When Merging Eight Models

Figure 10 shows 30 models sampled from the learned Pareto set by AdaMerging + TIES + Pareto Merging when merging eight ViT-B/32 models. We can see that Pareto Merging obtains a diverse model set. Most models dominate models obtained by the baselines, while none of the models is dominated by models obtained by baselines.

Figures 11 and 12 show 30 models sampled from the learned Pareto set by Task Arithmetic + Pareto Merging and TIES Merging + Pareto merging when merging eight ViT-B/32 models, respectively. We can see that Pareto Merging can achieve diverse trade-off around the base method. All of the models obtained by PM are non-dominated to each other in the 8-dimensional objective space. They are also non-dominated to the model obtained by the base method. Note that "dominance" observed in the 2D projections does not imply the real dominance in the 8-dimensional objective space. Unlike the base method, which offers only a single trade-off, PM allows users to select models tailored to their specific preferences.

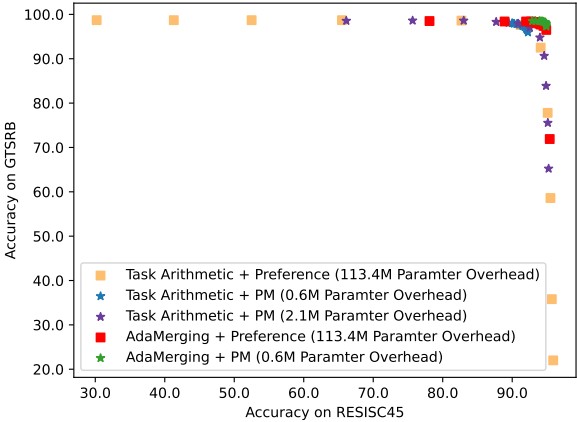

*Figure 9.* Full plot of accuracies of models obtained by two straightforward methods discussed Section 3.1 in when merging two ViT-B/32 models.

## E. Test accuracies when merging eight ViT-L/14 models

We further experiment on the larger ViT-L/14 backbone. The results are shown in Table 8. As can be seen, Pareto Merging still outperforms the baselines on most datasets, and the observations on ViT-B/32 model still hold.

*Table 8.* Test accuracies when merging eight ViT-L/14 models. Results on baselines are from (Yang et al., 2024b). The best result is in bold and the second best underlined.

| Method | SUN397 | Cars | RESISC45 | EuroSAT | SVHN | GTSRB | MNIST | DTD | Average |
|---|---|---|---|---|---|---|---|---|---|
| Individual | 82.3 | 92.4 | 97.4 | 100 | 98.1 | 99.2 | 99.7 | 84.1 | 94.2 |
| Traditional MTL | 80.8 | 90.6 | 96.3 | 96.3 | 97.6 | 99.1 | 99.6 | 84.4 | 93.5 |
| Weight Averaging | 72.1 | 81.6 | 82.6 | 91.9 | 78.2 | 70.7 | 97.1 | 62.8 | 79.6 |
| Fisher Merging | 69.2 | 88.6 | 87.5 | 93.5 | 80.6 | 74.8 | 93.3 | 70.0 | 82.2 |
| RegMean | 73.3 | 81.8 | 86.1 | 97.0 | 88.0 | 84.2 | 98.5 | 60.8 | 83.7 |
| DARE | 74.0 | 81.9 | 86.5 | 93.9 | 88.0 | 86.6 | 98.9 | 65.5 | 84.4 |
| MAP | 76.0 | 84.1 | 88.7 | 87.8 | 90.1 | 87.9 | 97.8 | 71.3 | 85.4 |
| Task Arithmetic | 73.9 | 82.1 | 86.6 | 94.1 | 87.9 | 86.7 | 98.9 | 65.6 | 84.5 |
| Task Arithmetic + PM (equal) | 74.0 | 82.1 | 86.7 | 94.1 | 87.9 | 86.8 | 98.9 | 65.7 | 84.5 |
| Task Arithmetic + PM (priority) | **74.3** | **82.5** | **87.3** | **95.0** | **91.0** | **88.1** | **99.1** | **66.1** | **85.4** |
| TIES Merging | 75.9 | **85.4** | 89.0 | 95.6 | 89.2 | 87.1 | 99.0 | 68.7 | 86.2 |
| TIES Merging + PM (equal) | 75.9 | **85.4** | 89.0 | 95.6 | 89.1 | 87.1 | 99.0 | 68.7 | 86.2 |
| TIES Merging + PM (priority) | **76.0** | **85.4** | **89.7** | **96.7** | **91.9** | **88.5** | **99.1** | **69.3** | **87.1** |
| AdaMerging | 79.0 | 90.3 | 90.8 | 96.2 | 93.4 | 98.0 | 99.0 | 79.9 | 90.8 |
| AdaMerging + PM (equal) | 79.8 | 90.7 | 91.4 | 96.5 | 93.5 | 98.3 | **99.1** | 80.6 | 91.2 |
| AdaMerging + PM (priority) | **80.5** | **91.7** | **91.9** | **98.1** | **96.4** | **98.8** | **99.1** | **80.8** | **92.1** |
| AdaMerging + TIES | 79.4 | 90.3 | 91.6 | 97.4 | 93.4 | 97.5 | 99.0 | 79.2 | 91.0 |
| AdaMerging + TIES + PM (equal) | 79.9 | 90.6 | 91.2 | 97.8 | 93.4 | 98.1 | 99.0 | 80.3 | 91.2 |
| AdaMerging + TIES + PM (priority) | **80.6** | **91.7** | **92.0** | **98.5** | **96.1** | **99.0** | **99.1** | **80.6** | **92.2** |

## F. Using Outer Product in $W(\gamma)$

In this experiment, we study other possibilities to transform the preference vector $\gamma$ to a $c \times d$ matrix using vector outer product (denoted as $\circ$):

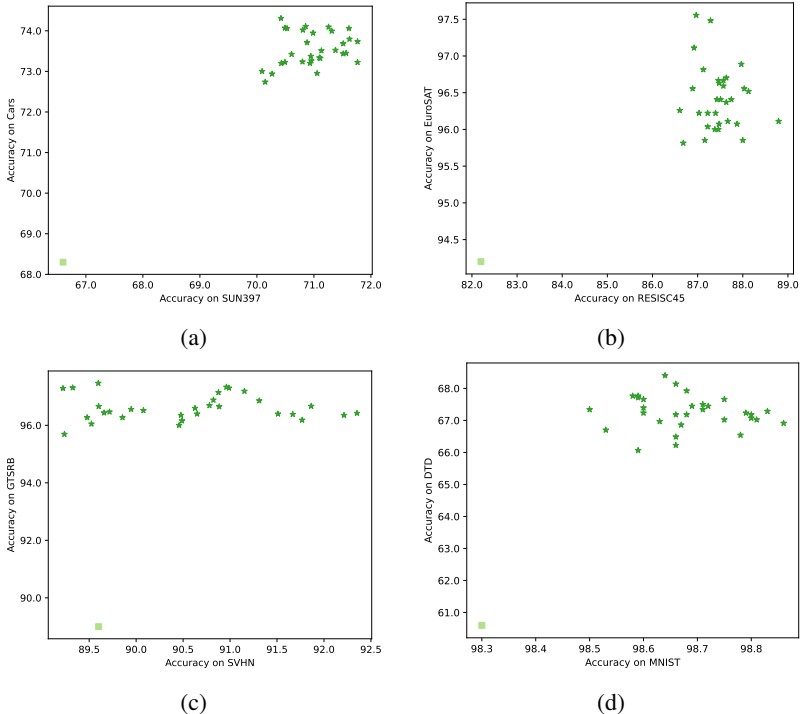

*Figure 10.* Models sampled from the learned Pareto set by AdaMerging + TIES + Pareto Merging when merging eight ViT-B/32 models. Each subplot shows the accuracies on 2 of the 8 tasks. For comparison, the square denotes the model obtained by AdaMerging + TIES.

- Variation 1: $W(\gamma) = v \circ A\gamma$, where $v \in \mathbb{R}^c$ and $A \in \mathbb{R}^{d \times K}$.

- Variation 2: $W(\gamma) = A\gamma \circ v$, where $v \in \mathbb{R}^d$ and $A \in \mathbb{R}^{c \times K}$.

The experimental settings are the same as in Section 4.3. The obtained results of merging two models fine-tuned on RESISC45 and GTSRB datasets are shown in Figure 13. We observe that both outer product-based method exhibit performance inferior to the proposed tensor-based method.

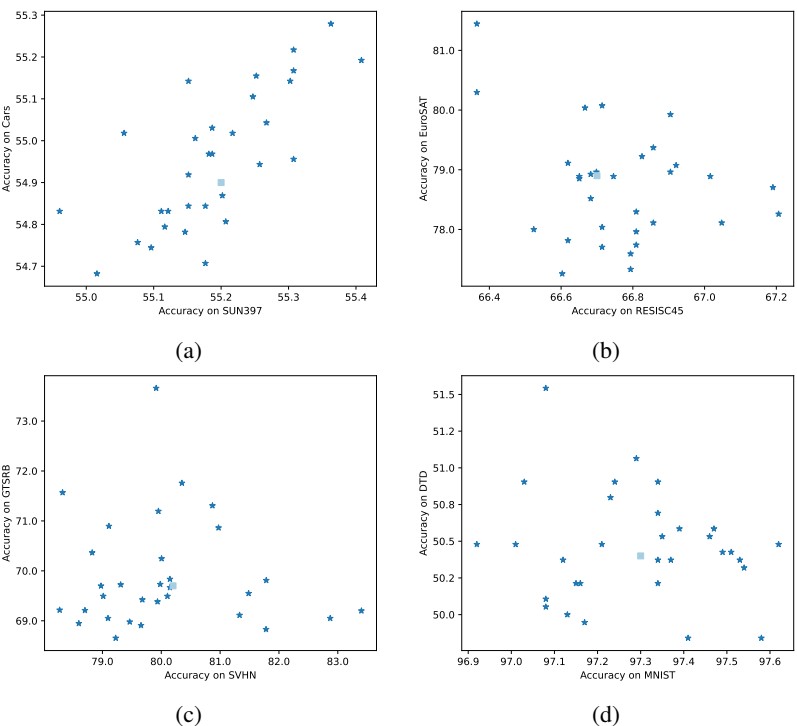

*Figure 11.* Models sampled from the learned Pareto set by Task Arithmetic + Pareto Merging when merging eight ViT-B/32 models. Each subplot shows the accuracies on 2 of the 8 tasks. For comparison, the square denotes the model obtained by Task Arithmetic.

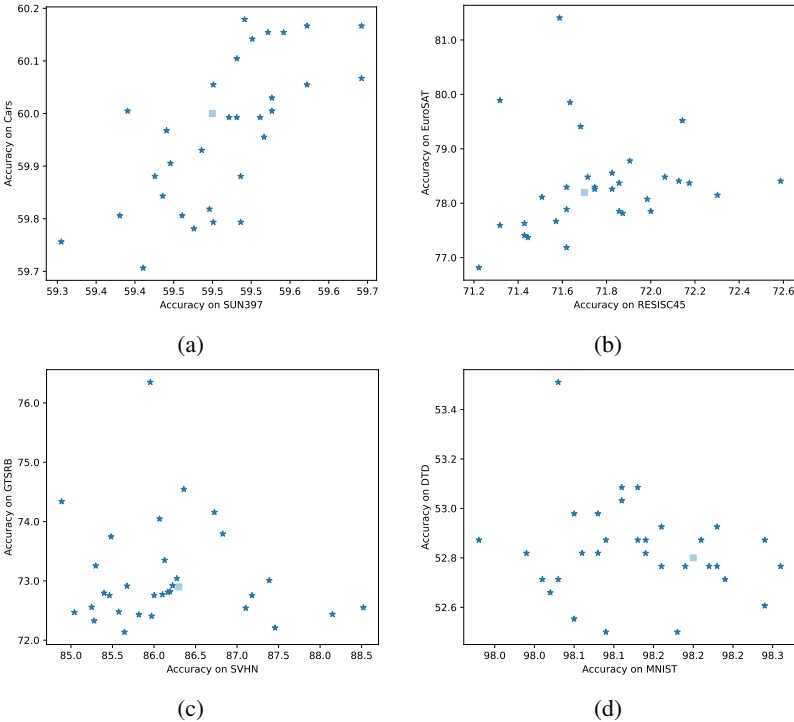

*Figure 12.* Models sampled from the learned Pareto set by TIES Merging + Pareto Merging when merging eight ViT-B/32 models. Each subplot shows the accuracies on 2 of the 8 tasks. The square denotes the model obtained by TIES Merging.

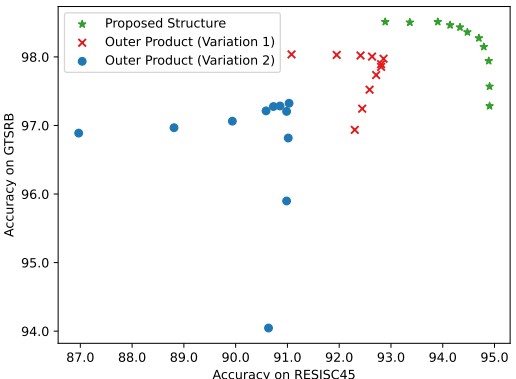

*Figure 13.* Comparison between the proposed tensor-based method and simple outer product-based method for preference-dependent personalization.

