# OpenReview forum: "Pareto Merging: Multi-Objective Optimization for Preference-Aware Model Merging"
_ICML.cc/2025/Conference — ICML 2025 poster_

### Official Review · Reviewer_6Su3 · 2025-03-01

**Overall Recommendation:** 2

**Summary:**

The authors use multi-objective optimization to create a framework for users to give preferences to which tasks are important to them when merging models. Given a merged model, they build another PEFT architecture that takes preference weights as input This new PEFT is trained either data-free or using unlabeled data. Experiments are done on toy examples and merging 8 image data sets.

**Claims And Evidence:**

The results do show improvements based on Pareto Merging, but I think some of what is demonstrated by Table 2 is problematic. It sticks out that Task Arithmetic and Task Artihmetic+PM(equal) perform almost identical. The weights in Task Arithmetic and other merging algorithms can also be selected to prioritize a single task in similar way that the preference vector is set for PM.

There is also missing insight. Why does AdaMerging+PM(equal) significantly outperform AdaMerging? The framework adds an additional PEFT module on top, so there is more complexity in the model. This also requires additional training and it is not clear how much training is needed. For an 8-dimensional preference vector that is sampled once per batch, it would seem that would require many epochs to sample the space of preference vectors. I didn't find any information about how many epochs are necessary in the Appendix.

An additional problem is that Table 6 in the Appendix using ViT-L/14 does not confirm the same significant outperformance claimed in Table 2. The main difference I see is that the individual models with ViT-L/14 significantly outperform the ViT-B/32 models (on average). And with better models, all merging models perform much better and the benefit of PM is not nearly as apparent. With a smaller benefit, the additional training requires more discussion.

**Essential References Not Discussed:**

No

**Experimental Designs Or Analyses:**

No issue

**Methods And Evaluation Criteria:**

Most evaluations make sense, but additional evaluations does Task Arithmetic and other merging methods by using the weights to assign priority would help see if the preference gives a benefit.

**Other Comments Or Suggestions:**

See above

**Other Strengths And Weaknesses:**

The main strength is that users can give preference to different models with very little overhead in terms of memory. But again, a weakness is that this could involve a lot of computational resources in terms of training. Furthermore, when considering priority, it is not clear when one would have priorities in image classification. If we have a priority in image classification, we would best served to use the individual model. The authors need to discuss when this is applicable in model merging for images. I can see where this would be useful for LLMs with model alignment but that is left for future work in the Conclusion. The approach with sampling preference vectors requires more discussion and analysis. What is the insight for optimizing the average loss with respect to preference vectors? See above for additional weaknesses in Claims and Evidence.

**Questions For Authors:**

See above

**Relation To Broader Scientific Literature:**

The contributions attempt to improve the state of the art in the literature.

**Theoretical Claims:**

NA

---

> ### Author Rebuttal · Authors · 2025-04-01
>
> Thank you for your valuable suggestions. Below, we provide a detailed response to each point.
>
> ---
> **R1: Direct incorporation of preference to Task Arithmetic and AdaMerging**
> In Section 3, we frame model merging as a MOO problem and propose both straightforward methods (Section 3.1) and our Pareto Merging (PM) approach (Sections 3.2-3.3). While straightforward methods incorporate preferences, they have significant limitations. Our PM method is more parameter-efficient and computationally effective, achieving better performance trade-offs.
>
> Following your suggestion, we tested Task Arithmetic + Preference and AdaMerging + Preference. Results (available at https://anonymous.4open.science/r/ICML-0464/) show:
>
> * Task Arithmetic + Preference has 113.4M parameter overhead and some extreme solutions obtained are typically undesirable for users. Our Task Arithmetic + PM approach with the default configuration (low-rank tensor applied only to attention layers) achieves reasonable trade-offs with 0.6M parameter overhead. Extending low-rank tensors to both attention and MLP layers shows better performance with only 2.1M parameters. Users can adjust settings to balance solution diversity.
>
> * AdaMerging + PM (0.6M parameter overhead) achieves better trade-offs than AdaMerging + Preference (113.4M parameter overhead). Moreover, our approach requires only a single run (0.28 GPU hours), whereas AdaMerging + Preference demands 11 runs (1.65 GPU hours).
>
> We will include the above disscussion in the final version.
>
> **R2: Task arithmetic and task arithmetic+PM(equal) perform almost identical, while AdaMerging+PM(equal) significantly outperform AdaMerging**
> Task arithmetic is data-free merging, so task arithmetic+PM(equal) has no additional information, resulting in similar performance. AdaMerging+PM leverages unlabeled data, allowing PM to better resolve task conflicts. This pattern of improvement with data incorporation appears consistently across various compression techniques, such as merging, pruning, and quantization.
>
> **R3: How much training is needed**
> We apologize for the omission of detail. We use 2,000 gradient steps. As shown in Table 1, the additional training time is small.
>
> **R3: ViT-L/14 results**
> The improvement needs to be considered with respect to traditional MTL, which serves as a performance upper bound. As can be seen from Table 6, with ViT-L/14, AdaMerging already achieves 90.8% (vs. 80.1% with ViT-B/32), naturally limiting improvement margins. Still, AdaMerging+PM (priority) achieves a significant 1.3% improvement over AdaMerging. This smaller margin reflects approaching optimal performance rather than method limitations.
>
> **R4: Computational overhead** As shown in Table 1, our computational overhead during training is small.
> for data-free merging (with task arithmetic), the overhead is 0.04 GPU hours; whereas for merging based on unlabeled data (with AdaMerging), the overhead is 0.13 GPU hours.
>
> **R5: Priorities in Image Classification and Extension to LLM**
> There seems to be a misunderstanding. Setting priorities does not mean that only one task is important while all other tasks are unimportant. Rather, it means that the model should excel at a specific task while still achieving acceptable performance on others. Maintaining separate models for different preferences is inefficient in terms of parameters, as it requires storing and managing multiple full models.
>
> Our Pareto Merging enables a single model to excel at prioritized tasks while maintaining reasonable performance across all tasks, with minimal parameter overhead. Additionally, our method supports flexible adjustment of prioritized tasks. We recognize the exciting potential of this approach for LLMs and are actively working on extending our algorithm to these models. However, due to time constraints, we can only include this extension in the final version of the paper.
>
> **R5: Preference Vector Sampling** Thanks for your question. We optimize the expected loss across the entire preference space to account for all possible user preferences. Different preferences weight the parameters in the low-rank tensor differently, enabling the model to capture diverse characteristics. While our method already demonstrates strong performance, we acknowledge that developing more effective preference vector sampling techniques is an interesting future work that could further enhance efficiency.
>
> ---
>
> We hope the above responses address your concerns. If you have any additional questions or suggestions, we are more than happy to discuss them further.

---

> > ### Comment · Reviewer_6Su3 · 2025-04-07
> >
> > Thank you for the rebuttal. I agree with some points such as R2 and R6. But a main sticking point is R5 which cannot be remedied without experiments. I do not see why this was done with the task of image classification in mind rather than for LLMs. I will maintain my score.

---

> > > ### Author Response · Authors · 2025-04-09
> > >
> > > Thank you for your reply. We are happy to know that some of your concerns are addressed. In response to your question on image classification, we have added a new experiment detailed below.
> > >
> > > First, recall that model merging is particularly useful in scenarios with limited computational resources, where the traditional approach of storing/running multiple task-specific models is impractical. In this new experiment, we consider three models that are independently fine-tuned on the GTSRB, RESISC45, and SVHN datasets, respectively.
> > >
> > > Suppose that a particular user has preference (or "priority" in our context) [GTSRB: 0.6, RESISC45: 0.2, SVHN: 0.2] (i.e., the user puts 60\% importance on the accuracy on GTSRB, and 20\% importance on each of RESISC45 and SVHN). Importantly, *priority does not mean ignoring lower-weighted tasks*. This setting is often encountered in the real world as users typically aim to obtain a **multi-task** model that prioritizes over certain tasks *while still achieving reasonably good performance on the other tasks*. For example, in resource-constrained environments such as edge devices and autonomous driving systems, different preferences emerge in varying contexts (e.g., different conditions, regions, users).
> > >
> > > The traditional approach does not use merging. One simply loads the three models and then use one corresponding to each image. However, this requires a large GPU memory (or frequent model loading/unloading which is also computationally expensive) and multiple independent forward passes.
> > >
> > > The following table compares the test accuracy of 4 methods:
> > >
> > > ||GTSRB(Weight 0.6)|RESISC45(Weight 0.2)|SVHN(Weight 0.2)|Weighted Average|
> > > |---|---|---|---|---|
> > > |Method 1: Single Model (GTSRB)|96.0|22.5|24.2|66.9|
> > > |Method 2: AdaMerging|91.8|96.3|93.8|93.1|
> > > |Method 3: AdaMerging+Preference|93.2|95.8|93.4|93.8|
> > > |Method 4: AdaMerging+PM|93.8|96.3|94.0|94.3|
> > >
> > > * Method 1, "use the individual model" as you suggested: In our example scenario, as the GTSRB task has the highest priority, we use the single-task model finetuned on GTSRB. As can be seen, while its performance on GTSRB is very good, it performs poorly on the remaining two tasks. This results in a poor weighted average of 66.9\%.
> > >
> > > * Method 2, AdaMerging: A representative model merging method. While it achieves good performance across all tasks, it does not consider the user preference, resulting in a suboptimal weighted average.
> > >
> > > * Method 3, AdaMerging + Preference: This is the naive extension introduced in Section 3.1 to incorporate user preference. Compared to AdaMerging, it improves the accuracy on GTSRB by 1.4\%, demonstrating that the proposed multi-objective optimization formulation is useful. However, as mentioned in Section 3.1 and R1, when there are $n$ user preference, this approach (i) requires $n$ optimization runs; and (ii) storage of all the three original models (226.8M parameter overhead) .
> > >
> > > * Method 4, the proposed PM approach: It shows significant improvement on the priority task of GTSRB (2\% over AdaMerging) while still maintaining good performance on the other tasks. Importantly, it efficiently handles multiple user preferences in a single optimization run with minimal parameter overhead (0.6M), addressing the key limitations of Method 3.
> > >
> > > Furthermore, as you mentioned, our approach can be easily extended to LLM alignment. The experiment
> > > is currently in progress. However, as finetuning on different alignment objectives to obtain models for merging is time-consuming, we will only be able to show the results in the final version.
> > >
> > > Overall, we believe the proposed preference-aware merging is important for both image and language models.
> > >
> > > We hope this clarification addresses your concerns. We deeply appreciate your thoughtful feedback and the time you have devoted to reviewing our paper. We will incorporate all your valuable suggestions in the final version.
> > >
> > > We would be very grateful if you would consider revising your assessment based on these clarifications. Thank you very much for your consideration.

---

### Official Review · Reviewer_FTzt · 2025-03-14

**Overall Recommendation:** 4

**Summary:**

This paper proposes a novel preference-aware multi-objective model merging method called Pareto Merging (PM) to generate a Pareto set of merged models (the number of models might be infinite) by a single merging process. The main contributions include 1) the preference-aware multi-objective model merging formulation and 2) the LoRA-based personalized model generation method with minimal parameter overhead. Experimental results show the proposed Pareto merging method can achieve state-of-the-art performance on different model merging problems with ViT-B/32 models.


**##After Rebuttal Comment##**

Thank you for your detailed response and new results. Since all my concerns have been adequately addressed, I raise my rating to 4.

**Claims And Evidence:**

Most claims in this work are well supported by clear and convincing evidence.

However, I have some concerns about the claim that "to the best of our knowledge, we are the first to utilize gradient-based MOO for model merging." To my understanding, there is a closely related work [1] that also investigates gradient-based MOO for model merging. A detailed discussion and comparison with [1] is required. In addition, this claim should be modified if needed.

[1] Towards Efficient Pareto Set Approximation via Mixture of Experts Based Model Fusion, arXiv:2406.09770.

**Essential References Not Discussed:**

As mentioned in the previous section, a closely related work on multi-objective model merging [1] is not discussed in this work.

**Experimental Designs Or Analyses:**

I've checked the experimental designs and analyses, and believe most of them are reasonable. I have the following concerns about the experiments.

1. As mentioned in the above sections, a comparison with the closely related work [1] could be helpful.

2. This work uses the smooth Tchebycheff scalarization for multi-objective optimization. It is interesting to know whether this method can truly outperform the simple linear scalarization and the original Tchebycheff scalarization in the model merging task.

**Methods And Evaluation Criteria:**

I believe the proposed method and the evaluation criteria both make sense for the multi-objective model merging problem. I have the following concerns about the proposed method.

1. To my understanding, the proposed Pareto Merging method has a similar model structure to the previous work on LoRA-based Pareto manifold learning [2, 3] (see Figure 2). The key difference is that PM uses a model merging approach to find the preference-independent based model, while previous work learns the base model. The pros/cons between these two approaches are not very clear to me. A detailed discussion could be very helpful to highlight the unique advantage of the proposed Pareto Merging approach.

2. The motivation of this work is to use the Pareto Merging method to find different models for different users rather than a single preference-conditioned model to adjust the trade-off among different objectives in real time. To my understanding, the proposed preference-independent based model + preference-dependent low-rank tensor structure could be promising for real-time trade-off adjustment. However, if the goal is to find different models for different users, why not just use model merging to find the whole model for each user once the user's preference is known? In what situation will we need to find a set of models for different users in parallel?

[2] Efficient Pareto Manifold Learning with Low-Rank Structure, ICML 2024.

[3] Pareto Low-Rank Adapters: Efficient Multi-Task Learning with Preferences, ICLR 2025.

**Other Comments Or Suggestions:**

N/A

**Other Strengths And Weaknesses:**

Strengths:

1. This work is generally well-written and easy to follow.

2. Multi-objective model merging is important for many real-world applications, espeically those with large foundation model. This work is a timely contribution on an important research direction.

3. The proposed Pareto merging method achieves state-of-the-art performance on different experiments.

**Questions For Authors:**

Please address my concerns raised in the above comments.

**Relation To Broader Scientific Literature:**

The proposed Pareto merging method is a natural extension of the previous methods on learning the entire Pareto set by a single model. I believe this extension is meaningful and valuable since the model merging method could be very important for real-world applications, especially those with large foundation models.

**Theoretical Claims:**

There is no theoretical claim or proof in this work.

---

> ### Author Rebuttal · Authors · 2025-04-01
>
> Thank you for your valuable suggestions. Below, we provide a detailed response to each point.
>
> ---
>
> **Response to Claims And Evidence**
>
> **R1: Comparison with [1]** Note that [1] is indeed a concurrent work of ours but with different goal. [1] aims to improve the efficiency of general Pareto set learning algorithm. Instead of training from scratch, [1] employs task arithmetic to merge specific modules while keeping others (e.g., MLP layers or both the MLP and attention layers) separate. They then train a MOE router using labeled data to weight these unmerged components.  Note that when both MLP and attention layers are not merged, [1] needs to keep the $K$ models after training.
>
> On the other hand, our method focuses on improving model merging and formulates it as a MOO problem. This extension has not been studied before (including [1]). Unlike [1], our method merges all modules while incurring a small low-rank tensor (with only 0.5\% parameters of the pretrained model).
>
> Note that formulating model merging as a MOO problem requires first identifying the MOO objectives for model merging, which is not trivial. For instance, in data-free merging, we observe that Task Arithmetic can be viewed as optimizing the distances between models in the weight space. This then allows us to define the MOO objectives based on these distances. While traditional model merging considers finding a single solution based on a specific preference, our MOO formulation transforms the problem to the finding of a continuous space of solutions, which is novel.
>
> To further compare with MoE-based fusion in [1], below we compare its performance on merging eight ViT-B/32 models using the same setup as in [1]. As reported in [1], with the use of labeled data, MoE-based fusion (with a final model size of 567M) achieves an average accuracy of 77.2\% when only the MLP is unmerged, and 83.5\% when all modules are unmerged (with final model size 1.02B). On the other hand, our method (with only 114M parameters and NOT requiring labeled data) achieves a higher accuracy at 85.2\%. We will add the above discussion to the final version.
>
> **Response to Methods And Evaluation Criteria**
>
> **R2: Comparison with LoRA-based Pareto Manifold leanring [2, 3]**  Thanks for your comment but that is not the key difference. As mentioned earlier, our primary objective is to improve model merging, rather than using model merging for efficient Pareto set learning. Our first key contribution is identifying the limitations of current model merging techniques and reformulating it as a MOO problem to effectively incorporate user preferences.  Furthermore, we propose an efficient preference-aware tensor structure, which is different from the weighted sum of LoRAs used in [2, 3].
>
> To further illustrate our advantages, we adapt [2,3] for use in our model merging setting. Using the setup in Section 4.2.2 and Table 2, the results on merging eight ViT-B/32 models are:
>
> |Method|Structure|Test Accuracy|Parameter Overhead|
> |---|---|---|---|
> | AdaMerging+PM (equal)|ours|84.9|0.61M|
> | AdaMerging+PM (equal)|structure in [2,3]|84.5|4.71M|
> | AdaMerging+PM (priority)|ours|85.5|0.61M|
> | AdaMerging+PM (priority)|structure in [2,3]|85.2|4.71M|
>
> As can be seen, the proposed tensor structure has better performance while significantly reducing the number of parameters. We will add the above discussion to the final version.
>
> **R3: Real-time trade-off adjustment & Merge for each preference independently** The proposed method can be used in both scenarios: (1) As you mentioned, it is promising for real-time trade-off adjustment. (2) When different users have different preferences, the straightforward preference incorporation method, as discussed in Section 3.1, has two limitations: (i) It requires storing all $K$ original models to address $n$ different user preferences. In contrast, our approach only requires a single model and a small low-rank tensor. (ii) For methods like AdaMerging, they require $n$ separate runs for $n$ preferences. For instance, handling 100 preferences with straightforward method would require $0.15 \times 100 = 15$ GPU hours, whereas our method requires only $0.28$ GPU hours.
>
> **Response to Experimental Designs Or Analyses**
>
> For comparison with [1], please refer to **R1**.
>
> **R4: Comparison with other scalarization methods** Thanks for your question. Following your suggestion, we add experiment with different scalarization methods on merging eight ViT-B/32 models (as in Section 4.2.2 and Table 2). The average accuracies obtained are: (i) linear scalarization: 84.3\%; (ii) Tchebycheff scalarization: 82.8\%; and (iii) smooth Tchebycheff scalarization: 85.2\%. These indicate that smooth Tchebycheff scalarization outperforms the other methods for our model merging task. We will add the discussion to the final version.
>
> ---
>
> We hope the above responses address your concerns. If you have any additional suggestions, we are more than happy to discuss them further.

---

> > ### Comment · Reviewer_FTzt · 2025-04-03
> >
> > Thank you for your detailed response and new results. Since all my concerns have been adequately addressed, I raise my rating to 4.

---

> > > ### Author Response · Authors · 2025-04-03
> > >
> > > We sincerely appreciate your thoughtful feedback and recognition of our work. We will incorporate all your suggestions into the final version. Thank you for your time and effort!

---

### Official Review · Reviewer_i5h5 · 2025-03-14

**Overall Recommendation:** 4

**Summary:**

This paper introduces a new method for merging models with trade-offs by finding the Pareto front. They included both data-free version and using unlabeled data version of the method.

**Claims And Evidence:**

Most of the claims are clear. I've detailed unclear points in the following comments.

**Essential References Not Discussed:**

I did not identify any essential references not discussed.

**Experimental Designs Or Analyses:**

Yes, they are standard baselines in the model merging literature, and the paper compared with various suitable baseline methods.

**Methods And Evaluation Criteria:**

Yes, they are standard baselines in the model merging literature, and the paper compared with various suitable baseline methods.

**Other Comments Or Suggestions:**

NA

**Other Strengths And Weaknesses:**

Strengths:
- A new algorithm for identifying pareto fronts for model merging.
- Compared with various existing baselines.
- Included both data-free and unlabeled data versions of the method.
Weakness:
- A few minor errors as I pointed out in questions.

**Questions For Authors:**

- In Table 1, how is the # parameters defined? I'm not sure why rewarded soup and MAP would double the parameter counts. Could you please clarify?
- In Table 1, the authors claim that the number of models MAP can generate is 100, because the initial population is 100. I don't think this is true because MAP uses NSGA III and the number of points on the final Pareto front can far exceed the initial population size.

**Relation To Broader Scientific Literature:**

This paper is relevant to the model merging community and for preference-aware model merging. Prior work in the field include MAP (model merging with amortized pareto front), and methods in MTL, such as ParetoMTL.

**Theoretical Claims:**

- There is no theoretical claims.

---

> ### Author Rebuttal · Authors · 2025-04-01
>
> Thank you for your valuable suggestions. Below, we provide a detailed response to each point.
>
> ---
>
> **Response to Questions For Authors**
>
> **R1: Parameter count in Table 1**
> The "parameter count" refers to the number of parameters that need to be stored after merging.
> Note that for each merging method, there are two options on what to store. For instance, with two models in the soup and 100 user preferences, one can either store the 100 merged models corresponding to these 100 preferences, or store only the two original models and merge them on-the-fly based on the user preference. Obviously, the latter is more memory-efficient, and is we adopt when calculating the parameter count. Hence, for MAP and the Rewarded Soups, the parameter counts are doubled in the two-model case.
>
> **R2: Number of solutions in Table 1**
> Thank you for pointing this out. In the paper, we mentioned 100 as the original NSGA-III algorithm enforces a fixed population size by removing dominated or overcrowded solutions during evolution. Upon double checking the MAP implementation, we observed that unlike the original NSGA-III, it retains all the non-dominated solutions, resulting in around 500 solutions at the end. We will correct this in the final version. However, please note that since we used the official implementation from the authors' github in the experiments,
> this does not affect any experimental results or discussions in the paper.
>
> ---
>
> We hope the above responses address your concerns. If you have any additional questions or suggestions, we are more than happy to discuss them further.

---

### Official Review · Reviewer_Pu7q · 2025-03-18

**Overall Recommendation:** 3

**Summary:**

The paper introduces a novel method named "Pareto Merging," which is designed for the efficient merging of multiple pre-trained machine learning models into a single model, taking into account the preferences of different users. The approach learns a set of models, each optimized for different trade-offs or preferences among the objectives, thereby offering customized solutions for varied user preferences.

**Claims And Evidence:**

Yes.

**Essential References Not Discussed:**

- The reference for MGDA is not accurate. Check the earlier work below.

[0] "Steepest descent methods for multicriteria optimization". Mathematical methods of operations research 2000.

- It would be better if the authors could discuss the relation with some other gradient-based preference-based multi-objective optimization works, as listed below. For example, how does the preference in this paper differ from the existing works? What are the pros and cons of using low-rank embedding for the preference?

[1] "A multi-objective/multi-task learning framework induced by Pareto stationarity" ICML, 2022.

[2] "FERERO: A flexible framework for preference-guided multi-objective learning" NeurIPS 2024.

[3] "PMGDA: A preference-based multiple gradient descent algorithm" arXiv:2402.09492, 2024.

- A detailed comparison should be made to some existing works on Pareto set learning or preference-conditioned models, some examples are listed below. Based on the discussion in Section 2.2, it seems that the only difference compared to the prior works for preference-conditioned models is that the authors "explore model merging for large models".

[4] "Smooth Tchebycheff scalarization for multi-objective optimization" ICML 2024.

[5] "Pareto set learning for expensive multi-objective optimization" NeurIPS 2022.

[6] "Panacea: Pareto Alignment via Preference Adaptation for LLMs" NeurIPS 2024.

**Ethical Review Concerns:**

No.

**Experimental Designs Or Analyses:**

No issues.

**Methods And Evaluation Criteria:**

Yes.

**Other Comments Or Suggestions:**

No.

**Other Strengths And Weaknesses:**

**Strengths**

1. The paper is well-written, and the method is clearly described.

2.  The proposed method is validated through some experiments.


**Weaknesses**

1. Does Table 1 provide the training or inference time? The authors should discuss the computational complexity in both training and inference and how it compares to some baselines.

2. I do not see a clear novelty compared to the reference [6] in **Essential References Not Discussed** section. This paper also seems to use a low-rank embedding for the preference. Same as [6], this paper can also generate infinite number of models.

**Questions For Authors:**

1. Based on the discussion in Section 2.2, it seems that the only difference compared to the prior works for preference-conditioned models is that the authors "explore model merging for large models" while the prior works focus on smaller models. However, the reference [6] in **Essential References Not Discussed** section is very similar to the approach proposed in this paper and also applies to large models.
How is this work different from [6]?

**Relation To Broader Scientific Literature:**

For model merging, this work considers preference-aware merging. The learned model with low-rank embedding can generate a Pareto set of merged models, with each representing a Pareto-optimal solution for a preference.

**Theoretical Claims:**

N/A

---

> ### Author Rebuttal · Authors · 2025-04-01
>
> Thank you for your valuable suggestions. Below, we provide a detailed response to each point.
>
> ---
> **Response to Essential References Not Discussed**
>
> **R1: MGDA Reference** Thanks. We'll include it in our final version.
>
> **R2: Relation with preference-based MOO works [1,2,3]** They differ fundamentally from ours. They aim to generate a single solution per run based on a given preference. To handle different preferences, they require multiple runs and obtain multiple models. On the other hand, our method produces a continuous Pareto set of solutions. Once trained, a MOO solution can be generated from the given preference without optimization. Currently, we use smooth Tchebycheff scalarization for its simplicity and effectiveness. But algorithms in [1,2,3] can be integrated into our framework by modifying Equation (10). Specifically, instead of directly using $\gamma_k$ to weight $S_k$, we can use algorithms in [1, 2, 3] to compute a weighting for $S_k$.
>
> **R3: Using low-rank embedding for the preference** There might be some misunderstanding. We are not using low-rank embedding for preference. Instead, we employ a low-rank tensor to integrate the preference into a single model, which is significantly more parameter- and compute-efficient compared to generating a separate model for each preference (using methods such as [1, 2, 3]) or use a full-rank tensor structure.
>
> **R4: Difference with Pareto set learning works [4, 5]**
> We would like to emphasize that our contributions are:
> 1. Problem formulation: We address the challenges of model merging by reformulating it as a MOO problem. This requires identifying the MOO objectives for model merging, and is not trivial. In data-free merging, we observe that Task Arithmetic can be viewed as optimizing the distances between models in the weight space. This then allows us to define the MOO objectives based on these distances.
> Moreover, while traditional model merging considers finding a single solution for a specific preference, our formulation transforms the problem to the finding of a continuous space of solutions, which is novel.
>
> 2. Scalable Pareto Set Learning: While works [4, 5] use hypernetworks (typically 100× larger than base networks), our low-rank tensor approach dramatically reduces the computational costs, making Pareto exploration feasible for large-scale models.
>
> **R5: Comparison with LLM alignment work [6]**
> Note that [6] is indeed a concurrent work with 3 key differences:
> 1. Problem Formulation:
> [6] is used for LLM alignment, while this paper is the first to apply MOO to model merging (both data-free and data-based). As mentioned in **R4**, it is non-trivial. As can be seen from empirical results, this leads to the ability
> to generate different models for different preferences
> and improved performance compared to the baseline algorithms.
> 2. Efficient Low-Rank Tensor Structure:
> [6] uses a SVD-LoRA-based approach, while we propose a low-rank tensor structure.  In the following, we adapt the approach in [6] for use in our model merging setting. We set the rank in their approach to 16, so that its number of parameters is comparable with ours.
> Using the setup in Section 4.2.2 and Table 2, the results on merging eight ViT-B/32 models are:
> |Method| Model|Test accuracy|
> |---|---|---|
> | AdaMerging+PM (equal)|ours | 84.9|
> | AdaMerging+PM (equal) | structure in [6]|84.1|
> | AdaMerging+PM (priority)| ours | 85.5|
> | AdaMerging+PM (priority) | structure in [6]|84.4|
>
> As can be seen, our low-rank tensor structure outperforms [6], particularly in the priority preference setting. This is because
> while [6] assigns a fixed singular matrix row to each objective, our tensor $G$ adaptively learns the relationships between objectives and so our structure is more flexible.
>
> 3. Optimization: [6] focuses on LLM alignment using labeled data, whereas we focus on model merging with either no data or unlabeled data. This distinction presents overfitting challenges, which we address using an efficient structure and tensor regularization.
>
> We will include the above discussions in the final version.
>
> **Response to Weakness 1**
>
> **R6: Training and inference times** Table 1 is on training time. As can be seen, our method has small computational overhead compared to the baselines. Specifically, during training, for a layer, AdaMerging optimization has a per-iteration complexity of $O(Kcd)$, while ours is $O(Kcd + (c+d)r + Kr^2)$ (where $K$ is the number of models, $c \times d$ the shape of the layer parameter, and $r$ is the rank.). As $r$ is much smaller than $c$ and $d$, the computational overhead is small.
>
> For inference, once the preference is fixed, the low-rank tensor can be merged into the base model, resulting in no inference overhead.
>
> **Response to Weakness 2 and Questions For Authors**
>
> Please refer to **R5**
>
> ---
> We hope the above responses address your concerns. If you have any additional questions or suggestions, we are more than happy to discuss them further.

---

### Decision · Program_Chairs · 2025-05-01

**Decision:**

Accept (poster)

**Comment:**

This paper proposes a preference-aware model merging based on multi-objective optimization problems. The experimental result for mering ViT models demonstrates that the proposed method can achieve better performance than the baseline methods. The paper is well-written, and the effectiveness of the proposed method is validated through experiments.

I would recommend that the authors include a discussion regarding related work discussed in the rebuttal. Also, it would be better to elaborate on the scope and target of the proposed method pointed out in the reviewer's comments and how to extend the proposed method to LL merging. I would also like to see the performance of the proposed method in merging LLMs if possible.

In summary, I think that this paper has a meaningful contribution to the community. I would recommend accepting this paper.